



# Assessing the Ducting Phenomenon and its Impact on GNSS Radio Occultation Refractivity Retrievals over the Northeast Pacific Ocean using Radiosondes and Global Reanalysis

Thomas E. Winning Jr. [1], Feiqin Xie[1] and Kevin J. Nelson[1,a]

[1] Texas A&M University – Corpus Christi, Corpus Christi, 78412, USA

[a] now at: Jet Propulsion Laboratory, California Institute of Technology, Pasadena, 91109, USA

*Correspondence to*: Thomas E. Winning Jr. (twinning@islander.tamucc.edu)

**Abstract.** In this study, high-resolution radiosondes from the MAGIC field campaign and ERA5 global reanalysis data are used to assess the elevated ducting layer characteristics along the transect over the northeastern Pacific Ocean from Los Angeles, California to Honolulu, Hawaii. The height of the planetary boundary layer (PBLH) increases as the strength of the refractivity gradient and resultant ducting decrease from east to west across the analysis transect. The thickness of the ducting layer remains remarkably consistent (~110 m) in the radiosonde data. On the other hand, the ERA5 generally resolves the ducting features well but underestimates the ducting height and strength especially over the trade cumulus region near Hawaii. A simple two-step end-to-end simulation is used to evaluate the impact of the elevated ducting layer on RO refractivity retrievals. A systematic negative refractivity bias (N-bias) below the ducting layer is observed throughout the transect, peaking approximately 70 meters below the PBL height (−5.42%), and gradually decreasing towards the surface (−0.5%). Further, the underestimation of the N-bias in the ERA5 data increases in magnitude westward and while the correlation of the N-bias with the minimum gradient and sharpness are all strong; there is no evidence of zonal dependence.

## 1 Introduction

The troposphere, where most weather occurs, consists of two main layers: the planetary boundary layer (PBL) and the free atmosphere (FA) (Garratt, 1992). The PBL characteristics change frequently on both spatial and temporal scales and the PBL height (PBLH) can impact the exchange of heat, momentum, and particulate matter with the FA, making it a critical factor in global energy balances and water cycling (Stull 1988; Ramanathan et al. 1989; Klein and Hartmann 1993). Regular PBL observations are mainly limited to in situ measurements from





surface stations and radiosondes. However, spatially and temporally dense in situ PBL
observations are only available from field campaigns such as the Boundary Layer Experiment
1996 (BLX96, Stull et al. 1997), the VAMOS Ocean-Cloud-Atmosphere-Land Study Regional
Experiment (VOCALS-REx, Wood et al. 2011), and the Marine Atmospheric Radiation
Measurement (ARM) GCSS Pacific Cross Section Intercomparison (GPCI) Investigation of
Clouds (MAGIC, Zhou et al. 2015), etc. Satellite observations of the PBL are also limited due to
signal attenuation of the conventional infrared sounder in the lower troposphere and the low
vertical resolution of microwave sounding instruments. Additionally, while the depth of the
PBLH can vary from a couple hundred meters to a few kilometers (von Engeln and Teixeira
2013; Ao et al. 2012), the transition layer from the PBL to the FA is typically on the order of tens
to hundreds of meters thick (Maddy and Barnet 2008), rendering ineffective PBL sensing from
the low vertical resolution passive infrared and microwave sounders.
On the other hand, Global Navigation Satellite System (GNSS) radio occultation (RO) provides
global atmospheric soundings with a vertical resolution of approximately 100 m in the lower
troposphere under all weather conditions (Kursinski et al. 1997, 2000). One of the major GNSS
RO missions is the Formosat-3/Constellation Observing System for Meteorology, Ionosphere,
and Climate (COSMIC), later referred to as COSMIC-1 (Anthes et al. 2008), and its follow-on
mission COSMIC-2 (Schreiner et al. 2020). Numerous studies have documented the high value
of GNSS RO for profiling the PBL and determining the PBLH (Nelson et al. 2021; Winning et
al. 2017; Ao et al. 2012; ; Guo et al. 2011; Basha and Ratnam 2009; Ao et al. 2008; Xie et al.

51 2008).

The advancement of the GNSS RO technique with open-loop tracking (Sokolovskiy et al., 2006;
Beyerle et al., 2003; Ao et al., 2003) along with the implementation of the radio-holographic
retrieval algorithm (Jensen et al., 2004; Jensen et al., 2003; Gorbunov, 2002) have led to much
improved PBL sounding quality. However, probing the marine PBL remains challenging as
systematic negative biases are frequently seen in RO refractivity retrievals (Feng et al. 2020; Xie
et al. 2010). One major cause of the refractivity bias (hereafter N-bias) is the RO retrieval error
due to elevated atmospheric ducting often seen near the PBLH (Ao et al., 2007; Xie et al., 2006;
Ao et al. 2003; Sokolovskiy 2003, ). This elevated ducting prevails over the subtropical eastern
oceans (von Englen et al., 2003; Lopez, 2009, Feng et al., 2020), and the horizontal extent of
ducting in these regions can be on the order of thousands of kilometers (Winning et al. 2017; Xie





et al. 2010). In the presence of ducting, the vertical refractivity gradient exceeds the critical
refraction threshold for L-band frequencies (i.e., d$N$/dz $\leq -157$ N-units km$^{-1}$). The steep negative
refractivity gradient is often observed in the vicinity of the PBLH, which is typically caused by
an atmospheric temperature inversion, a moisture lapse, or a combination of both. When ducting
is present, the Abel inversion in the standard retrieval process encounters a non-unique inversion
problem due to a singularity in the bending angle, resulting in large, systematic underestimation
of refractivity ($N$) below the ducting layer (Sokolovskiy, 2003; Ao et al., 2003; Xie et al. 2006).
The large uncertainty in RO refractivity coupled with the singularity in bending angle hinders
assimilation of RO observations into numerical weather models, resulting in discarding of a
significant percentage of RO measurements inside the PBL (Healy, 2001).
In order to thoroughly evaluate the N-bias attributed to ducting, the issue must be examined from
the ground up by using a dense collection of observations where the occurrence of ducting in the
lower troposphere is present in the daily climatology of the region. Section 2 provides details of
the two data sets used for this study: high-resolution radiosondes over the northeastern Pacific
Ocean and ERA5 reanalysis profiles colocated to the radiosondes. Additionally, we discuss the
method used for colocation between the radiosondes and ERA5 profiles, as well as detection of
the ducting layer and the corresponding PBLH. Section 3 presents the ducting climatology for
key variables, such as ducting height, PBLH, minimum N-gradient, and gradient sharpness. The
characteristics of ducting including the thickness and strength along the cross-section are also
shown. Furthermore, we evaluate the ducting-induced N-bias in GNSS RO refractivity retrievals
by carrying out a two-step end-to-end simulation. Section 4 summarizes the findings and
discusses the direction of future research.
## 2      Data and methods
### 2.1 MAGIC radiosonde and colocated ERA5 data sets
A collection of high-resolution radiosondes from the Marine Atmospheric Radiation
Measurement (ARM) GCSS Pacific Cross Section Intercomparison (GPCI) Investigation of
Clouds (MAGIC) are utilized as the primary data set in this analysis (Lewis 2016; Zhou et al.
2015). The MAGIC field campaign took place from 26 September 2012 to 2 October 2013 as
part of the U.S Department of Energy ARM Program Mobile Facility 2 (AMF2) aboard the





Horizon Lines container ship, *Spirit*, which completed 20 round trip passes between Los
Angeles, California and Honolulu, Hawaii during the yearlong data collection period (Painemal
et al., 2015; Zhou, 2015). During each transit, radiosondes were launched at 6-hour intervals
from the beginning of the program through the end of June 2013; the observation frequency
increased to every 3 hours from July 2013 through the end of the campaign (Zhou et al., 2015).
A total of 583 MAGIC radiosonde profiles were collected during the field campaign (Zhou et al.,
2015), all with a vertical sampling frequency of 0.5 Hz (2 seconds), which provides an average
vertical sampling interval of ~8 m below 3 km.
The number of observations and location (Fig. 1) of this data set serves multiple benefits. First,
the northeast Pacific transitions from a shallow stratocumulus-topped PBL to a higher, trade-
cumulus boundary layer regime along the GPCI transect (Garratt, 1992); this unique transition
zone provides an ideal natural laboratory for studying the horizontal variation of the marine PBL.
Second, the large number of observations over a 12-month time frame provides high temporal
(diurnal and seasonal) and spatial profiling of the PBL along the GPCI transect. Finally, ducting
is prevalent throughout the domain over which the observations were captured which creates an
opportunity to perform an analysis over a natural cross-section of X (zonal) and Z (vertical)
dimensions.

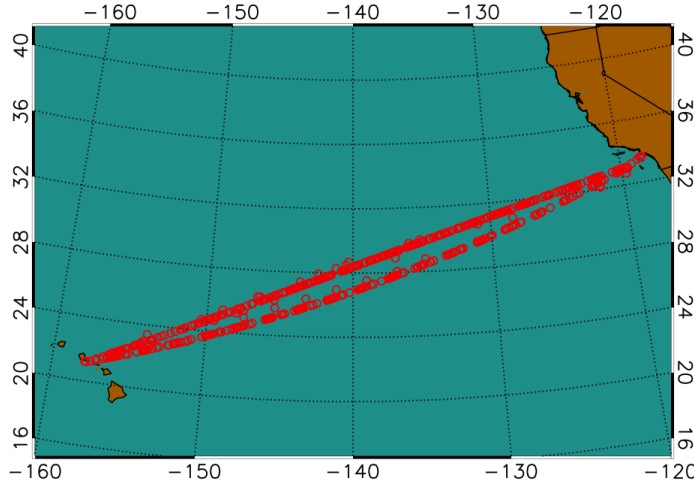

**Figure 1: Location of radiosonde observations from the MAGIC field campaign October 2012–September 2013.**

The radiosonde profiles are colocated with the ECMWF Reanalysis version 5 (ERA5, Hersbach
et al. 2020). The ERA5 reanalysis data have a horizontal grid resolution of 0.25°x0.25°, 1-hour
temporal resolution, and 137 vertical levels from the surface to 0.01 hPa (Hersbach et al., 2020).





114 An average of 19 model levels exist below 1 km providing the highest vertical resolution near

115 the surface; vertical density of the model decreases with height to 8 levels within the 1 km–2 km

116 layer and further decreasing to 5 levels within the 2 km–3 km. Each MAGIC radiosonde profile

117 was colocated with the nearest ERA5 grid point that is within 1.5 hours of the closest 3-hourly

118 model reanalysis profile.

119 **2.2 PBL height detection with the minimum gradient method**

120 At GNSS L-band frequencies, the atmospheric refractivity ($N$ in N-units) is derived from the

121 refractive index $n$, where $N= (n-1) \times 10^6$ and, in the neutral atmosphere (Kursinski et al., 1997),

122 is a function of the atmospheric pressure ($P$ in mb), temperature ($T$ in K), and partial pressure of

123 water vapor ($P_w$ in mb) as seen in Eq. (1) from Smith and Weintraub (1953).

124 $N = 77.6\frac{P}{T} + 3.73 \times 10^5 \frac{P_w}{T^2},$      (1)

125 Atmospheric refractivity decreases exponentially with height which, all else being equal yields a

126 negative value vertical gradient. As such, the minimum refractivity describes the largest

127 magnitude value.

128 Over the subtropical eastern oceans, a sharp decrease in moisture is often associated with a

129 strong temperature inversion marking a clear transition from the PBL to the FA. Both the

130 moisture lapse and the temperature inversion lead to a sharp negative refractivity gradient which

131 can be precisely detected from GNSS RO. Numerous studies have implemented the simple

132 minimum gradient method to detect the PBLH, which is the location of the minimum refractivity

133 gradient (Ao et al., 2012; Seidal et al., 2010; Xie et al., 2006). When the vertical refractivity

134 gradient is less than the critical refraction (dN/dz $\approx$ −157.0 N-units km$^{-1}$), ducting occurs

135 (Sokolovskiy, 2003). To better assess the strength of the refractivity gradient for more robust

136 PBLH detection with gradient method, Ao et al. (2012) introduced the sharpness parameter,

137 which is defined as the ratio of the minimum vertical refractivity gradient to the root mean

138 square error of the refractivity gradient profile (eq. 2).

139 $\tilde{X}' \equiv -\frac{X'_{min}}{X'_{RMS}},$      (2)

140 Each refractivity gradient profile can then be filtered to identify the PBLH values with sharpness

141 parameter exceeding a specific threshold, thus increasing the robustness of PBLH detection. In

142 this study, the MAGIC radiosonde refractivity profiles were first interpolated to a uniform 10 m





vertical grid and then smoothed by 100 m to reduce the noise in the N-gradient profile that is a
result of the high sampling rate. Colocated ERA5 data were also vertically interpolated to the
same 10 m grid but not smoothed as these data do not contain the inherent noise as the
radiosonde observations.
**2.3 Ducting layers**
Instances of multiple ducting layers occurring within a profile are present for both the MAGIC
(31.5%) and ERA5 (6.7%) data sets. A ducting layer is identified as any interval of continuous
points with refractivity gradient equal to or less than $-157$ N-units km$^{-1}$. Note, however, we only
refer to the "ducting layer" of each profile as the dominant ducting layer corresponding to the
layer in which the minimum gradient is located (Fig. 2a–d). The ducting layer thickness ($\Delta h$) is
defined as the interval between the top and bottom of the ducting layer where the N-gradients
reach critical refraction. Similarly, the strength of each ducting layer ($\Delta N$) is defined as the
refractivity difference between the bottom and top of the ducting layer. The ducting layer height
is in reference to the top of the ducting layer (Ao, 2007), which is generally slightly above the
PBLH.
Figure 2 illustrates two ducting layers in a representative MAGIC radiosonde case near $-150°$,
but only one in the colocated ERA5 profile. Profiles of radiosonde refractivity (N-units x 1/10,
$N/10$), temperature ($T$) and specific humidity ($q$) and their respective gradients (d$N$/dz, d$T$/dz and
d$q$/dz) are shown in Fig 2a and Fig. 2b, respectively. Similar plots for the collocated ERA5
profiles are shown in Fig. 2c and Fig. 2d. The PBLH of the radiosonde (2.10 km) is almost
identical to the colocated ERA5 (2.14 km) and the "dominant" ducting layer near the PBLH
demonstrates similar thickness. However, a second, weaker ducting layer seen in the radiosonde
above the PBLH was not captured by the ERA5. This is likely due to the lower vertical
resolution in ERA5 as can be seen in the gradient plots (Fig. 2b and Fig. 2d).
It is also worth noting that the residual layer between 1.2–1.5 km with gradient close to critical
refraction is seen in the radiosonde is also seen in the ERA5 profile, but at a much lower altitude
(~0.7 km).

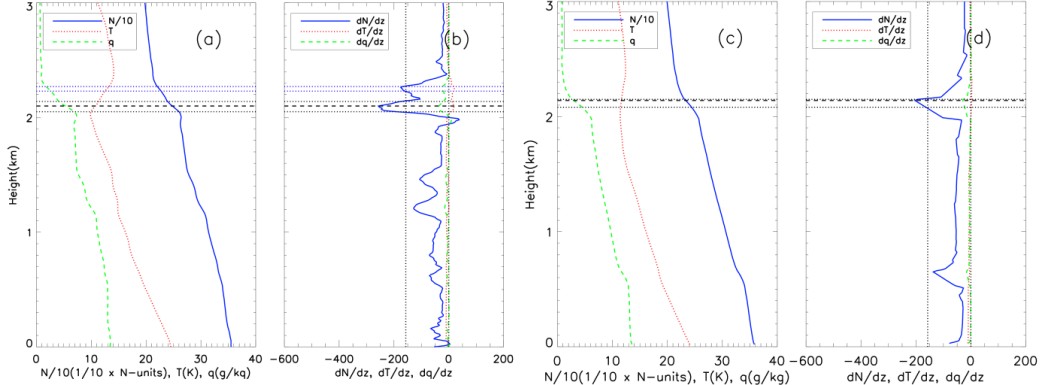

**Figure 2: (a) MAGIC radiosonde (−150.00˚) and (c) colocated ERA5 (−150.00˚) profiles of refractivity (1/10 x N-units, $N/10$, solid blue), temperature ($T$ in K, dotted red) and specific humidity ($q$ in g kg$^{-1}$, dashed green); (b) the associated radiosonde and (d) ERA5 gradient profiles. The horizontal dashed line highlights the height of the minimum gradient, i.e., PBLH. The paired horizontal dotted lines represent the bottom and top of the two ducting layers in the radiosonde profile, but only one in the ERA5 profile.**

## 2.4 Evaluation of GNSS RO N-bias resulting from ducting

In order to estimate the systematic negative N-bias in GNSS RO observations in the presence of ducting, we use an end-to-end simulation on the radiosonde and ERA5 refractivity profiles. The simulation consists of a two-step process adapted from Xie et al. (2006). The first step is to simulate the 1-dimentional GNSS RO bending angle as a function of impact parameter by forward Abel integration of an input refractivity profile assuming a spherically symmetric atmosphere. The second step is to simulate the GNSS RO refractivity retrieval by applying the Abel inversion on the simulated bending angle from step one. In the absence of ducting, the impact parameter (i.e., the product of refractive index and the radius of the curvature) decreases monotonically with height, allowing a unique solution to the inverse Abel retrieval. However, in the presence of an elevated ducting layer, the Abel retrieval systematically underestimates the refractivity profile due to the non-unique Abel inversion problem resulting from the singularity in bending angle across the ducting layer (Sokolovskiy 2003; Xie et al., 2006). It should be noted that an additional 50 m vertical smoothing has been applied to the simulated bending angle profiles of both radiosonde and ERA5 data sets to alleviate the challenge of integration through the very sharp bending angle resulting from ducting in the inverse Abel integration procedure (Feng et al., 2020).

Figure 3 shows the end-to-end simulation results for the same radiosonde (a–d) and the colocated ERA5 (e–h) cases from Fig. 2. Figures 3a and 3e show the input refractivity profile ($N_{rds}$ and



$N_{ERA5}$) and corresponding Abel refractivity retrieval ($N_{Abel}$), respectively. The PBLH is marked
by a horizontal dotted line. The peak bending angle is consistent with the sharp refractivity
gradient. Figure 3b shows the fractional N-bias between the simulated Abel retrieved RO
refractivity profile and the observation, i.e., (($N_{Abel} - N_{Obs}$)/$N_{Obs}$). Considering the significant
spatial and temporal variations of ducting height along the transect, each N-bias profile is
normalized to its PBLH for the purposes of comparison. For example, the zero-adjusted height
refers to the PBLH for each individual profile. The systematic negative N-bias is clearly shown
below the ducting layer marked by the PBLH in both cases, with the biases decreasing at lower
altitude, the largest magnitude bias (−5% for radiosonde; −2.5% for ERA5) close to the ducting
height and a minimum magnitude approaching zero near the surface.

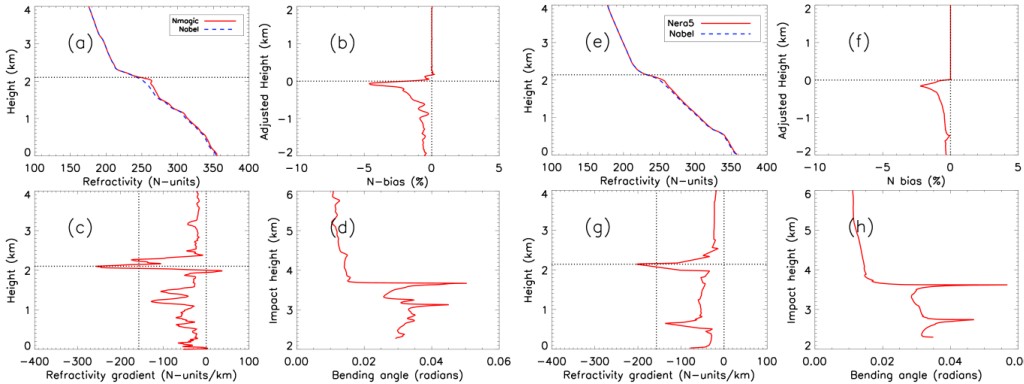


**Figure 3: Four-panel comparison of individual profiles of $N_{Obs}$ vs. $N_{Abel}$ that are reconstructed through the end-to-end**
**simulation. Four-panels for MAGIC of: (a) $N_{Obs}$ (solid red) and $N_{Abel}$ (blue dashed) from surface to 10 km; (b) adjusted N-**
**bias (($N_{Abel} - N_{Obs}$)/ $N_{Obs}$); (c) minimum gradient and (d) bending angle vs. impact parameter. Colocated ERA5 profiles**
**are shown in panels e-h, respectively.**
## 3    Analysis
Out of a total of 583 MAGIC radiosonde (and co-located ERA5) profiles, quality control has
been implemented based on five key criteria. First, a total of 19 radiosonde and 24 ERA5 profiles
near the southern California coast were removed due to a zonal position east of −120° or
anomalously high PBL heights (PBLH > 3.0 km) with no distinct minimum gradient. The
remaining profiles in the easternmost portion of the domain were too few in number to calculate
meaningful statistics. Second, any profile lacking critical refraction (i.e. dN/dz < −157 N-units
km$^{-1}$) points was excluded from the analysis which resulted in the removal of 47 radiosonde and
176 ERA5 profiles. Third, the noisy bending angle could result in errors in Abel refractivity


retrieval and cause positive N-bias. Therefore the profiles with N-bias greater than +0.5% are
excluded resulting in the removal of 61 MAGIC profiles and 16 ERA5 profiles. Fourth, the
profiles with only surface ducting are discarded when the only refractivity gradient less than
$-157$ N-units $km^{-1}$ occurs below the 300 m threshold. Finally, 25 radiosonde profiles and 2
ERA5 profiles were removed due to the Abel retrieval failure. After implementing all quality
control measures, the number of radiosonde and ERA5 profiles used for the N-bias analysis is
reduced to 396 and 319 profiles, respectively.
**3.1 PBL climatology**
To evaluate the ducting climatology along the transect from the coast of southern California to
Hawaii, we group the MAGIC radiosonde and the colocated ERA5 profiles into eight 5°
longitude bins between $-160.0°$ and $-120.0°$. The equally spaced bins are centered at $-157.5°$,
$-152.5°$, $-147.5°$, $-142.5°$, $-137.5°$, $-132.5°$, $-127.5°$ and $-122.5°$ which allows for the spatial
variation of the PBL, ducting layer and the associated properties along the transect to be easily
illustrated. Figure 4 shows the median value of PBLH (a), sharpness (b) and minimum gradient
(c) along the transect. The median-absolute-deviation (MAD) for each parameter is also shown.
In Fig. 4a, the MAGIC radiosondes clearly show the gradual increase of the PBLH along the
transect from the shallow stratocumulus-topped PBL (~800 m) near the southern California coast
westward to the much deeper trade-cumulus regime (~1.8 km) near Hawaii. A similar structure is
seen in the colocated ERA5 data but with an average low bias of 165 m below the radiosonde.
However, a nearly 800 m underestimation in PBLH over the two westernmost bins near Hawaii
is also seen, this is consistent with what is found over the equivalent trade cumulus region of the
subtropical southeast Pacific Ocean (Xie et al., 2012). Such a discrepancy could be due to the
decreasing vertical resolution with height in the ERA5 profiles. This results in a sharper
refractivity gradient caused by the frequent residual layer (below 1 km) as compared to the actual
PBLH near 2 km. Note that the larger median absolute deviation for the westernmost bins
compared to the rest of the transect illustrates the existence of greater PBLH variability closer to
the trade-cumulus boundary layer regime.
The westward decreasing magnitude of the minimum refractivity gradient (Fig. 4b) and
sharpness parameter (Fig. 4c) indicates the westward weakening of moisture lapse and/or





temperature inversion across the PBL top, which is consistent with the decreasing synoptic-scale
subsidence from the California coast to Hawaii.

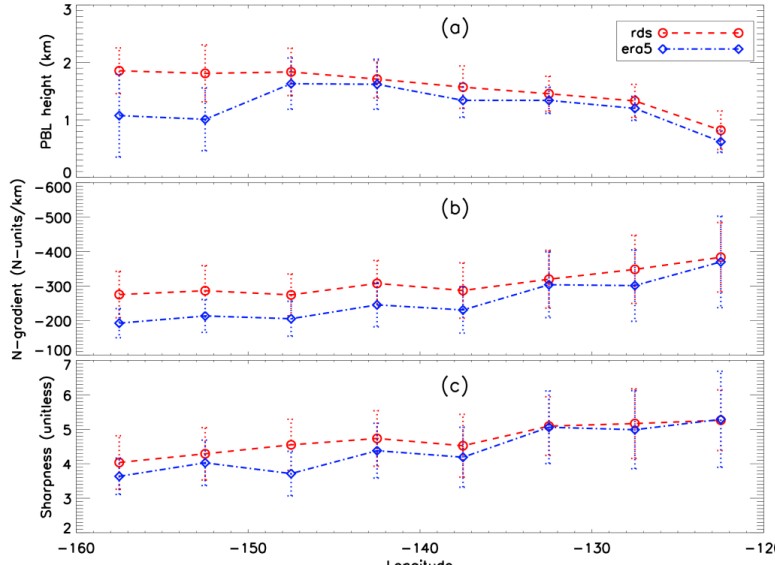


**Figure 4: Zonal transect of 5° bin MAGIC and ERA5 (a) PBLH, (b) sharpness parameter and (c) minimum refractivity gradient for MAGIC (median in red circle, MAD in dashed error bars) and ERA5 (median in blue diamond, MAD in dot-dashed error bars).**


It is also notable that the ERA5 systematically underestimates not only the PBLH, but also the
magnitude of the minimum N-gradient across the entire transect; this can also be seen in the
sharpness parameter west of −132.5°. This discrepancy could again be partially attributed to the
decrease in vertical sampling in ERA5 profiles as compared to the radiosondes, the result of
which leads to a weaker PBL N-gradient and coincides with an increasing PBLH. Therefore, the
underestimation of the ERA5 minimum N-gradient increases in magnitude from east to west and
becomes most prominent near Hawaii where the PBLH reaches the maximum height over the
region.
**3.2 Ducting climatology**
As introduced in Sect. 2.3, the key characteristics of the ducting layer along the transect will be
investigated, these include the ducting layer height, thickness ($\Delta h$), and strength ($\Delta N$), as well as
the average refractivity gradient within the ducting layer ($\Delta N/\Delta h$).





The ducting layer heights from both radiosonde and ERA5 show a westward increase along the
transect (Fig. 5a), which is similar to the PBLH in Fig. 4a. Note again that the ERA5 shows a
systematic ~100–200 m low bias when compared to the radiosondes between −122.5˚ and
−147.5˚, with the difference increasing to more than 500 m near Hawaii.
The ducting layer thickness is the median height from the bottom of the ducting layer to the top
and is expressed in km (Fig. 5b). Ducting thickness ($\Delta h$) for MAGIC shows a near constant
value of 110 m across the entire transect with only a slight increase to 130 m at −122.5˚; this is
consistent with findings from Ao et al. (2003). Conversely, the ERA5 shows a constant but
slightly thicker ducting layer to the east of −137.5˚ and then a decreasing thickness to the west of
−137.5˚ (Fig. 5b).
The ducting layer strength is the decrease in refractivity from the bottom of the ducting layer to
the top (Fig. 5c) and the ratio $\Delta N/\Delta h$ reflects the average gradient of the ducting layer (Fig. 5d).
The ducting strength ($\Delta N$) for the radiosondes ranges from 25 N-units near Hawaii to 40 N-units
near the coast of California. Both $\Delta N$ and $\Delta N/\Delta h$ show an overall westward decreasing trend
along the transect which is consistent with the decrease in magnitude of the N-gradient (Fig. 4b).
Note that MAGIC and ERA5 show similar ducting strength in the eastern part of the region but
diverge near −137.5˚ with ERA5 10 to 20 N-units weaker than the MAGIC profiles. On the other
hand, ERA5 shows a systematic lower average refractivity gradient ($\Delta N/\Delta h$) than MAGIC
throughout the transect, indicating the challenge in ERA5 to consistently resolve the sharp
vertical structure in refractivity, and likewise in temperature and moisture profiles, across such a
thin ducting layer. The problem becomes acutely clear near the trade cumulus region.



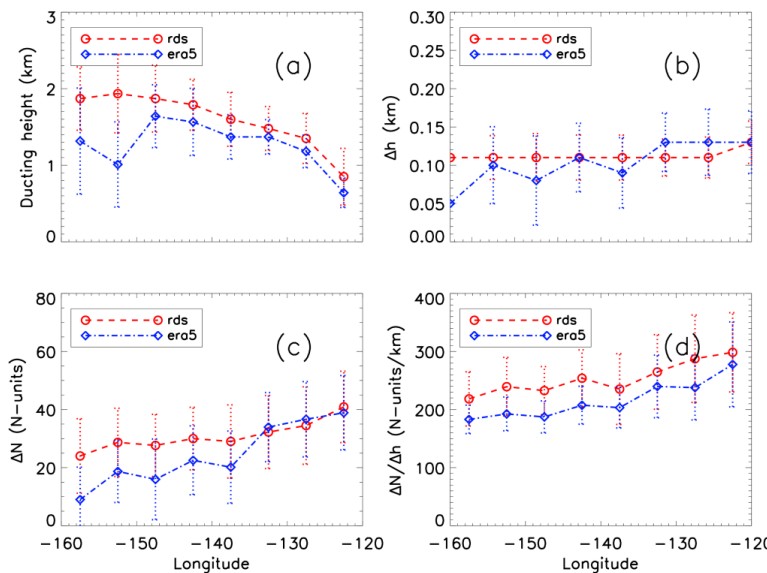

**Figure 5: Zonal transect of 5° bin median (a) ducting height, (b) ducting layer thickness (Δh), (c) ducting layer strength (ΔN), and (d) average ducting layer gradient ΔN/Δh for MAGIC (median in red circle and red-dashed line, MAD in red-dotted error bars) and ERA5 (median in blue diamond and dot-dashed error bars, MAD in blue-dotted error bar).**

Figure 6 shows ducting layer thickness as a function of ducting layer strength, with each data point colored by its respective longitude bin. The relationship between $\Delta h$ and $\Delta N$ is not longitude-dependent for either data set, but a linear trend is evident for thinner ducting layers ($\Delta h < 0.1$ km) with weaker ducting strength ($\Delta N < \sim25$ N-units). However, for the ducting layers thicker than the median value of 0.1 km, such a trend becomes less identifiable, and the ducting strength $\Delta N$ begins to show more variability toward larger values.



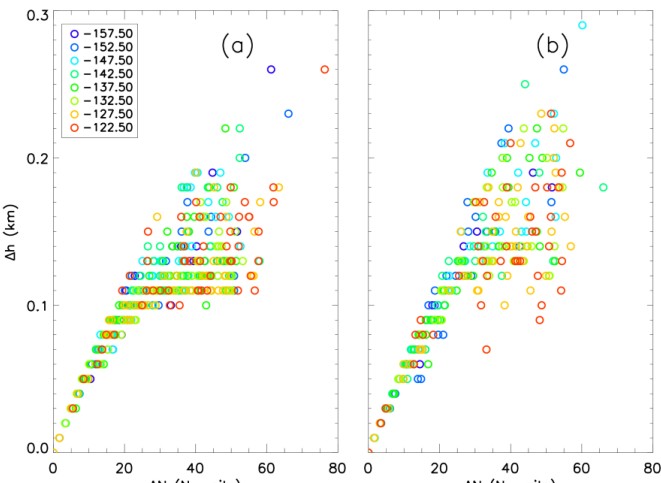

**Figure 6: Comparison of individual profiles' ducting strength (ΔN) vs. ducting thickness (Δh) for MAGIC (a) and ERA5 (b). The color of each circle represents the location of the 5° longitude bin of each observation.**

### 3.3 Ducting-induced GNSS RO N-bias statistics

To estimate the systematic negative N-bias in GNSS RO observations due to ducting, we have applied the end-to-end simulation to all radiosonde and ERA5 refractivity profiles with at least one elevated ducting layer detected (details in Sect. 2.5). The N-bias climatology along the transect as well as its relationship to the ducting properties are presented below.

### 3.3.1 N-bias climatology

Figure 7 shows a composite of both MAGIC (396 profiles) and ERA5 (319 profiles) N-bias profiles which have been normalized to their PBLH, with the median N-bias and MAD overlaid. The comparison reveals a number of occurrences of multiple ducting layers above the minimum gradient identified PBL in the MAGIC data while there are significantly less occurrences in the ERA5 data. Figure 7 illustrates the systematically negative N-bias peaks at nearly 100 m below the PBLH (ducting height) and decreases at lower altitudes. Many radiosonde profiles show smaller negative N-biases above the PBLH (e.g., zero adjusted height), but only a few in ERA5 which is a result of the secondary ducting layers above the major ducting layer near PBLH. The peak median value of the N-bias for radiosondes is −5.42% (MAD, 2.92%), nearly twice the ERA5 value of −2.96% (MAD, 2.59%). It is worth noting that the variabilities (MAD) between the radiosonde and ERA5 data are very close to each other.



A closer look at each data set reveals that the difference between the 5° median PBLH and height
of the maximum N-bias ($h_{PBL} - h_{N\text{-bias}}$) is positive for all bins. The maximum difference of 100 m
is located in bin −137.5° and a minimum difference of ~15 m at bin −152.5°. Comparatively, the
ERA5 reflects a PBL height greater than the N-bias height for each bin with a maximum
difference of 230 m located at −142.5° and a minimum of ~45 m at −157.5°. The ERA5 data
show a larger average height difference between the PBL and N-bias (120 m) than the
radiosonde data (70 m).
The N-bias comparison of the 5° bin median values of the two data sets favors the radiosonde
data with smallest magnitude difference located at bin −147.5° (−4.37%) and largest magnitude
difference of −7.86% located at bin −122.5°. Comparatively, the ERA5 minimum N-bias
difference of −0.77% −157.5°) is much lower than the radiosonde while the maximum difference
is similar in both magnitude −5.92%) and location (−122.5°).

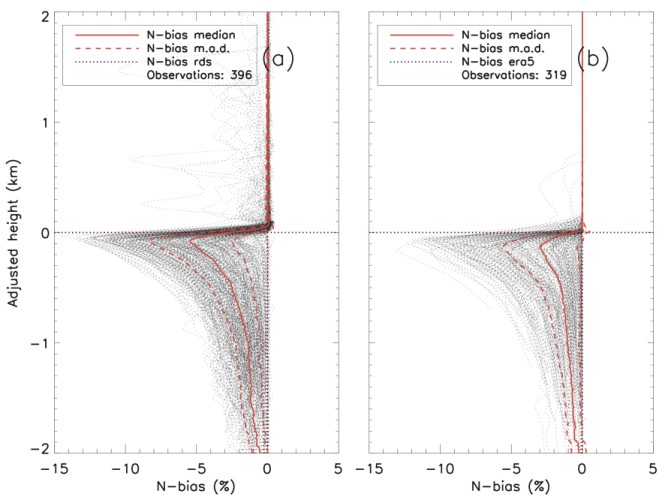

**Figure 7: Fractional refractivity difference (N-bias) between the simulated Abel-retrieved refractivity profile and the**
**original observation profile (($N_{Abel} - N_{Obs}$)/$N_{Obs}$), for all individual observations (dotted gray): (a) MAGIC radiosondes**
**(396 total profiles) and (b) ERA5 (319 total profiles) with population median (solid red) ± MAD (dashed red). Note the**
**zero value in the adjusted height refers to the PBLH for each individual N-bias profile.**
**3.3.2 N-bias along the transect**
To illustrate the large variation in the N-bias vertical structure resulting from the spatial variation
of ducting height and strength, we separately present the N-bias profiles (median ± MAD) for
each 5° bin, replacing the zero adjusted height with the median PBLH for each bin (Fig. 8). The
radiosonde composite (Fig. 8a) illustrates the transition of the median N-bias height from 1.8 km





at Honolulu, HI to 0.8 km near the coast of Los Angeles, CA. Table 1 provides supplemental

values for the Fig. 8 illustration of the radiosonde and ERA5 statistical climatology. The

radiosonde N-bias variation shows a minimum magnitude of near the center of the transect and

two of the largest magnitude difference values of as the bookends while the ERA5 N-bias values

have a larger range but peak values (−5.41% to −6.23%) in the three bins closest to California;

note the significantly reduced peak N-bias to the west of −137.5˚ (−3.10% to −0.71%).

Moreover, a discontinuity exists in the two westernmost longitude bins (−157.5˚ and −152.5˚)

which show a markedly lower and weaker N-bias.

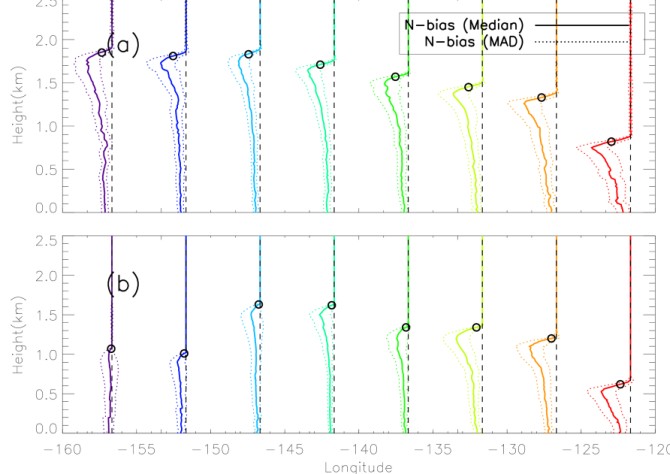

**Figure 8: Median N-bias (solid) ± MAD (dotted) along the north Pacific transect for MAGIC radiosondes (a) and ERA5 (b). Open circles represent the median PBL height for each 5˚ bin.**

**Table 1: 5˚ bin median and MAD peak N-bias values for MAGIC radiosondes (RDS) and ERA5.**

| Peak N-bias | | | | |
|---|---|---|---|---|
| Longitude | RDS median | RDS MAD | ERA5 median | ERA5 MAD |
| -157.5˚ | -6.11 | ±2.85 | -0.71 | ±1.80 |
| -152.5˚ | -5.24 | ±2.91 | -2.23 | ±1.68 |
| -147.5˚ | -4.85 | ±2.18 | -2.03 | ±2.25 |
| -142.5˚ | -5.78 | ±2.44 | -3.10 | ±2.24 |
| -137.5˚ | -5.34 | ±2.95 | -2.60 | ±2.21 |
| -132.5˚ | -5.92 | ±3.14 | -5.41 | ±2.79 |
| -127.5˚ | -6.42 | ±3.38 | -5.60 | ±2.74 |
| -122.5˚ | -8.10 | ±3.27 | -6.23 | ±2.98 |





Figure 9 further illustrates the peak N-bias, median PBL N-bias and the near surface N-bias (at
0.3 km) at each bin along the transect. Note that the quality control process removes the
refractivity profiles below 0.3 km. Therefore, the median N-bias is the median value from the
near surface (0.3 km) to the PBLH.
Contrary to the general trend of westward decrease in magnitude of the minimum N-gradient
(Fig. 4b) and ducting strength (Fig. 5c), the radiosonde peak N-bias shows the maximum
(median: −8.10%, MAD: 3.26%) near California (−122.5˚) and the minimum (median: −4.85%,
MAD: 2.18%) over the transition region (−147.5˚) as well as a slight increase to a secondary
maximum (median: −6.11%, MAD: 2.85%) near Hawaii (−157.5˚). The median PBL N-bias and
the near surface N-bias also show a similar pattern. However, the median N-bias demonstrates a
sharp decrease in the eastern half of the domain from −5.25% (MAD: 2.71%) at −122.5˚ to
−1.71% (MAD: 1.26%) at −137.5˚, and then remains relatively constant over the western half of
the domain. Similarly, the near surface N-bias reaches a maximum magnitude of −3.54% (MAD:
2.11%) and sharply decreases to −1.06% (MAD: 0.85%) at −137.5˚, and then remains relatively
constant over the western half of the domain.
It is important to point out that the much higher ducting height and larger variation near Hawaii
as compared to California leads to smoothed and much smaller median N-gradient values (Fig.
4b), which also results in a smaller N-bias without being normalized to the PBLH. Therefore, the
normalized N-bias observed near Hawaii indicates the presence of strong ducting over the trade-
cumulus boundary layer regime (Fig. 8a), which will lead to comparable N-bias to that over the
stratocumulus topped PBL.
On the other hand, the ERA5 data show a westward decrease of all three N-biases, systematically
underestimating all three as compared to the radiosondes. This is expected as the decrease of
ERA5 vertical resolution at higher altitude leads to a weaker PBL N-gradient observation (Fig.
4b), and thus weaker ducting and a smaller ducting-induced N-bias. Such underestimation of the
N-bias in the ERA5 is at a minimum near California where the PBLH is lowest but becomes
more severe westward with an increase in height, reaching a maximum magnitude N-bias
difference near Hawaii. In this case, the peak N-bias is merely −0.71% (MAD: 1.80%) as
compared to −6.23% (MAD: 2.98%) at −122.5˚ (Fig. 9a and Table 1). The large difference seen
in the N-bias along the transect strongly indicates the challenges of the ERA5 data to resolve the
sharp gradient across the ducting layer, resulting in a large variation in PBLH of the ERA5 data



in the western segment of the region. The increasing difference between the radiosonde and
ERA5 data from east to west is most pronounced in the peak N-bias cross-section (Fig. 9a) but is
also clearly evident in both the median N-bias (Fig. 9b) as well as the near surface N-bias (Fig.
9c).

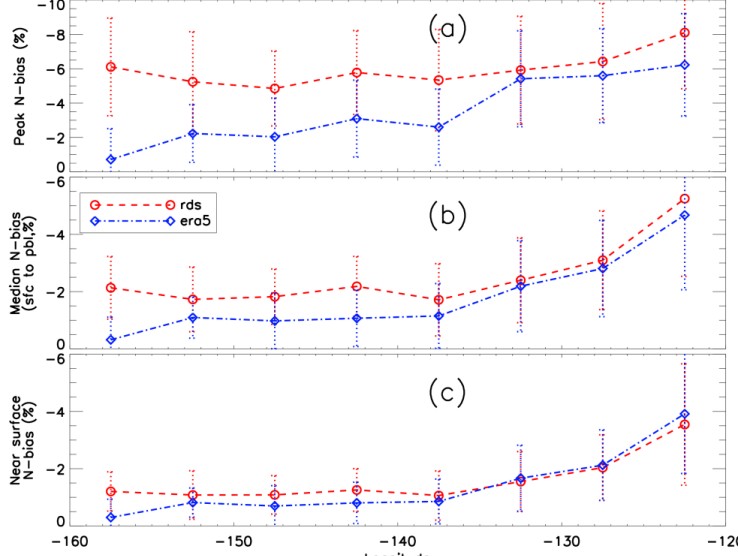

**Figure 9: Zonal transect of 5° bin (a) peak N-bias, (b) median PBL N-bias, and (c) near surface N-bias at 0.3 km for**
**MAGIC (median in red circle and red-dashed line, MAD in red-dotted error bar) and ERA5 (median in blue diamond**
**and dot-dashed line, MAD in blue-dotted error bar)**

### 3.3.3 N-bias climatology and key variable analysis

Figure 10 shows a scatter plot of the PBLH vs. height of maximum N-bias along the transect
with each data point colored by the center longitude of the bin to which it belongs. The PBLH
and the height of maximum N-bias show a clear linear relationship with high correlation for both
the MAGIC (0.89) and ERA5 (0.98) data. The majority of the radiosonde data show the heights
of the maximum N-bias aligns well with the PBLH but with a very small low bias (less than 70
m). The reason for the lower correlation value when compared to the ERA5 data is attributed to
the radiosonde N-bias profiles with a double peak at which the larger magnitude bias is located
(Fig. 7a). On the other hand, the ERA5 maximum ducting heights show little difference from the
PBLH near California (e.g., −122.5°), but become lower moving westward, which is illustrated
by the increasing difference between the linear regression line and the 1:1 line.





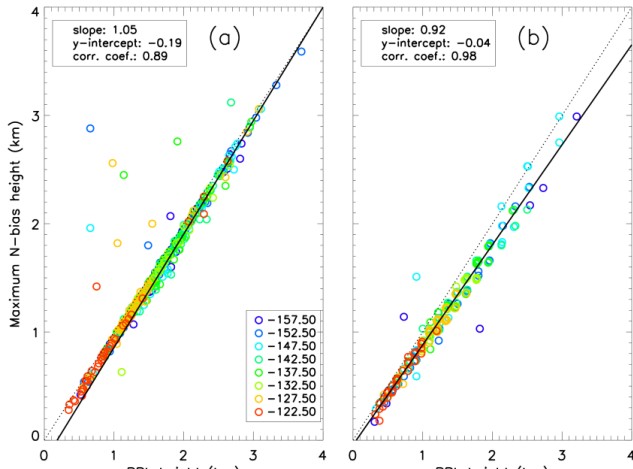

**Figure 10: PBLH vs. height of maximum N-bias for individual profiles from MAGIC (a) and ERA5 (b) data. The color of each open circle represents the center longitude of the 5° bin to which each profile belongs.**

Figure 11 shows the minimum refractivity gradient as a function of ducting-induced refractivity bias for MAGIC radiosondes (a) and ERA5 (b) and the corresponding sharpness parameters (c) and (d), respectively. A near-linear relationship between the minimum refractivity gradients and the maximum N-biases is evident for both MAGIC radiosondes and ERA5 profiles; in other words, the sharper the N-gradient, the larger the N-bias. The linear fit function along with the correlation coefficient for both MAGIC radiosondes (0.93) and the ERA5 profiles (0.88) are also presented.

The sharpness parameter (Fig. 11c, 11d) also shows a linear relationship with the maximum N-bias which is a result of its dependence on the minimum N-gradient. While a similar conclusion can be reached, it is interesting to note that the difference in the correlation of the radiosonde (−0.83) and the ERA5 (−0.84) does not lie in the observations with the larger magnitude peak N-bias, but in those closer to zero as the radiosonde data clearly centers below the regression line and trends above while the ERA5 with peak N-bias less than 5% are centered around the regression line. In the case of both key variables, their relationship with the peak N-bias exhibits no indication of zonal dependence.



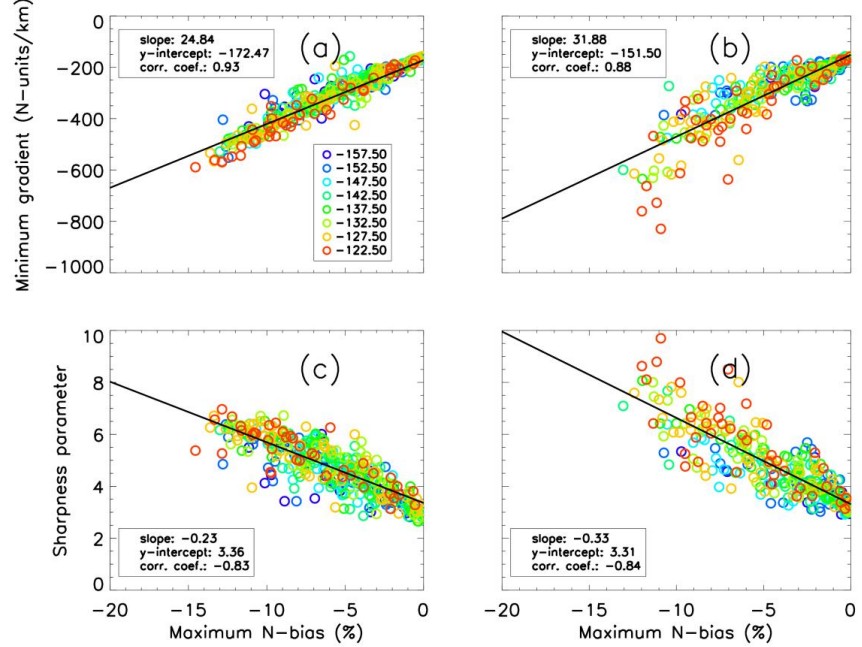

**Figure 11: (a, b) Minimum refractivity gradient (N-units km⁻¹) and (c, d) sharpness parameter, as a function of the**
**maximum N-bias (%) for MAGIC (a, c) and ERA5 (b, d) data with the line of linear regression in solid black. Color of**
**each open circle represents the center longitude of the 5° bin to which each profile belongs.**
4 **Summary and Conclusions**
In this study, radiosonde profiles from the MAGIC field campaign have been analyzed to
investigate the ducting climatology and the impact of associated systematic refractivity biases
that occur over the eastern North Pacific Ocean between Hawaii and California. Colocated ERA5
reanalysis data were used as a secondary comparison to the radiosonde observations.
The nearly 1-year high-resolution MAGIC radiosonde dataset reveals the frequent presence of
ducting at a well-defined PBL throughout the transect marked by a sharp refractivity gradient
resulting from the large moisture lapse across a strong temperature inversion layer. The PBLH
increases by more than 1 km along the transect from CA to HI while the magnitude of the N-
gradient decreases by 100 N-units km$^{-1}$. The zonal gradient of both variables illustrates the
transition of the PBL from shallow stratocumulus adjacent to the California coast to deeper
trade-wind cumulus that are prevalent near the Hawaiian Islands.
To estimate the systematic negative N-bias in GNSS RO observations due to ducting, we applied
an end-to-end simulation on all radiosonde refractivity profiles that contained at least one





elevated ducting layer. The ducting layer thickness remained remarkably consistent (110 m)
across the transect with westward decreasing strength and increasing height. The ERA5 slightly
underestimates both the height and strength of the ducting layer and so the PBLH.
The maximum N-bias occurs just below the PBLH, where the refractivity gradient is strongest.
The height of the maximum N-bias and the PBLH show a highly positive correlation. The mean
difference between the two is about 70 meters in the radiosonde but increasing to about 120
meters in the colocated ERA5 data. The correlation between the PBLH and the height of the
maximum N-bias is highly positive.
MAGIC radiosondes indicated larger values of both ducting strength ($\Delta N$) and thickness ($\Delta h$)
than from ERA5 in the western half of the transect. The reverse is true in the eastern portion of
the domain, and is likely associated with the transition of the cloud layer from open-cell cumulus
in the west to stratocumulus and stratus in the east (Wood et al., 2011; Bretherton et al., 2019).
While this segment of the transect also coincides with a better sampling rate for the ERA5 data
(~40 m vertical resolution), the ERA5 continues to systematically underestimate the average
ducting layer gradient climatology ($\Delta N/\Delta h$) when compared to the radiosondes. The largest N-
bias is located in the region of strongest ducting which also corresponds to the largest sharpness
parameter. The limited number of model levels in ERA5 near 2 km causes ducting to be
underrepresented near the trade wind inversion which is evident in the discrepancy between the
radiosonde and ERA5 PBLH cross sections.
Future work will include a comprehensive simulation study to explore the regional difference in
horizontal inhomogeneity and its impact on GNSS RO soundings. This research will improve
RO data quality, enhance understanding of PBL inhomogeneity, and advances weather and
climate prediction capabilities.









## 5 Data availability

Data for the Marine Atmospheric Radiation Measurement (ARM) GCSS Pacific Cross Section Intercomparison (GPCI) Investigation of Clouds (MAGIC, Zhou et al., 2015) can be accessed through the U.S. Department of Energy's Office of Science https://www.arm.gov/research/campaigns/amf2012magic.

Data for the ECMWF Reanalysis version 5 (ERA5, Hersbach et al., 2020) can be accessed at https://www.ecmwf.int/en/forecasts/dataset/ecmwf-reanalysis-v5.

## 6 Author contribution

Author Thomas Winning is responsible for all original text and, data analysis and production of graphics. Author Kevin Nelson contributed by providing updated data processing code, colocation of ERA5 data with MAGIC observations and first and second round edits. Author Feiqin Xie is the academic advisor for the primary author and also provided draft edits and paper organization and writing guidance.

## 7 Acknowledgements

The authors acknowledge funding support of earlier work from NASA grant (NNX15AQ17G). Authors T. Winning and K. Nelson were also partially supported by research assistantship from Coastal Marine System Science Program at Texas A&M University – Corpus Christi. The high-resolution ERA5 reanalysis data were acquired from ECMWF. The MAGIC radiosonde data were provided by the Atmospheric Radiation Measurement program (ARM) Climate Research Facility sponsored by the U.S. Department of Energy (DOE).

Author T. Winning's current affiliation: Ventura County Air Pollution Control District, Ventura, CA, 93003, USA. Author T. Winning acknowledges this work was done as an academic pursuit in association with Texas A&M University – Corpus Christi and not in the author's capacity as an employee of the Ventura County Air Pollution Control District.

Author K. Nelson's current affiliation: Jet Propulsion Laboratory, California Institute of Technology, Pasadena, 91109, USA. Author K. Nelson acknowledges this work was done as a private venture and not in the author's capacity as an employee of the Jet Propulsion Laboratory, California Institute of Technology.



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
