# Peer review of "Assessing the Ducting Phenomenon and its Potential Impact on"

_Atmospheric Measurement Techniques, 2023_

## Referee Comment (RC1)

Review of Assessing the Ducting Phenomenon and the Impact on GNSS Radio Occultation

Refractivity over Northeast Pacific Ocean using Radiosondes and Global Analysis

Recommendation: Accept with minor revisions.

This work is original and important. The authors compare the characteristics of PBL height and

ducting phenomena along the transect from Los Angeles to Hawaii using radiosondes and ERA5

analysis. They also assess the impact of ducting on GNSS Radio Occultation Refractivity. I

only have a few minor comments.

Minor comments:

1.   L22-23: Delete "there is no evidence of zonal dependence".

2.   L33: What does VAMO stand for?

3.   L35: What does GCSS stand for?

4.   L66: Add (Fjeldbo et al. 1971) after "inversion".

5.   L76: Replace "ERA5" with "the ECMWF Reanalysis version 5 (ERA5, Hersbach et al.
     2020)".

6.   L99-107: Shorten this paragraph focusing on the benefits of the data set for this study.

7.   L101: Replace "(Garret, 1992)" by "(Garret, 1994)".

8.   L111-112: Replace "the ECMWF Reanalysis version 5 (ERA5, Hersbach et al. 2020)" with
     "ERA5".

9.   L113: Remove "(Hersbach et al. 2020)".

10.  L143: The procedures for smoothing the data by 100 m should be described.

11.  L167: Explain what you mean by "the residue layer".

12. Figure 2: When was this sounding taken? The figure caption for (a), (b), (c), (d) is out of sequence.

13. L181: Add a reference to Abel integration.

14. L189: How did you perform 50 m vertical smoothing?

15. L247: What is the meaning of "moisture lapse"?

16. L249: Cite Riehl (1979) or any other references at the end of this sentence.

17. Section 3: It might be useful to plot a typical ERA5 sounding and compare it with the observed radiosonde sounding near Hawaii as well as off the California coast.

Reference:

Riehl, H. 1979: Climate and weather in the tropics. London: Academic Press. 611 pp.

---

## Author Comment (AC1)

The authors thank the anonymous reviewer for their comments. Please see the individual responses to the reviewer's comments below.

Reviewer #1 Comments – Minor Revisions

Review of Assessing the Ducting Phenomenon and the Impact on GNSS Radio Occultation Refractivity over Northeast Pacific Ocean using Radiosondes and Global Analysis

Recommendation: Accept with minor revisions.

This work is original and important. The authors compare the characteristics of PBL height and ducting phenomena along the transect from Los Angeles to Hawaii using radiosondes and ERA5 analysis. They also assess the impact of ducting on GNSS Radio Occultation Refractivity. I only have a few minor comments.

Minor comments:
1. L22-23: **Delete "there is no evidence of zonal dependence".**
   a. Deleted

2. L33: **What does VAMO stand for?**
   a. The official name for acronym Variability of the American Monsoon Systems (VAMOS) has been added to text.

3. L35: **What does GCSS stand for?**
   a. Thank you for bringing this to the authors' attention, the abbreviation GCSS is a collection of other abbreviations as is the acronym MAGIC. Each are listed below.
      i. Global Energy and Water Experiment (GEWEX)
      ii. GEWEX Cloud System Studies (GCSS)
      iii. ARM (Atmospheric Radiation Measurement)
      iv. In full, the MAGIC campaign abbreviation uses the first letter of the abbreviations of other field studies that also use the first letter of other abbreviations (Zhou et al., 2015).
      v. *M* (Marine) *A*(Atmospheric Radiation Measurement) *G* (GPCI (GEWEX (Global Energy and Water Experiment) Pacific Cross Section Intercomparison)) *I*nvestigation of *C*louds
   b. Full names listed in b and c above added to the text.
      i. '…field campaigns such as the Boundary Layer Experiment 1996 (BLX96, Stull et al. 1997) and the Variability of the American Monsoon Systems (VAMOS) Ocean-Cloud-Atmosphere-Land Study Regional Experiment (VOCALS-REx, Wood et al. 2011), and the Marine Atmospheric Radiation Measurement (ARM) Global Energy and Water Experiment (GEWEX) Cloud System Studies (GCSS) Pacific Cross Section Intercomparison (GPCI) Investigation of Clouds (MAGIC, Zhou et al. 2015).'

4. **L66: Add (Fjeldbo et al. 1971) after "inversion".**
   a. Citation added, "Fjeldbo, G., Kliore, A.J., and Eshleman, V.R.: The Neutral Atmosphere of Venus as Studied with the Mariner V Radio Occultation Experiment. *Astron. J.*, 76, 123-140, doi.org/10.1086/111096, 1971."

5. **L76: Replace "ERA5" with "the ECMWF Reanalysis version 5 (ERA5, Hersbach et al. 2020)".**
   a. Replaced

6. **L99-107: Shorten this paragraph focusing on the benefits of the data set for this study.**
   a. Removed two separate lines in order to streamline this paragraph
   b. Shortened to:
      i. 'Use of this data set serves multiple benefits. First, the northeast Pacific transitions from a shallow stratocumulus-topped PBL to a higher, trade-cumulus boundary layer regime along the GPCI transect (Garratt, 1994). Second, the large number of observations over a 12-month time frame provides high temporal (diurnal and seasonal) and spatial profiling of the PBL along the GPCI transect (Fig. 1). Finally, ducting is prevalent throughout the domain which creates a natural cross-section of X (zonal) and Z (vertical) dimensions.'

7. **L101: Replace "(Garret, 1992)" by "(Garret, 1994)".**
   a. Replaced

8. **L111-112: Replace "the ECMWF Reanalysis version 5 (ERA5, Hersbach et al. 2020)" with "ERA5".**
   a. Replaced

9. **L113: Remove "(Hersbach et al. 2020)".**
   a. Citation removed

10. **L143: The procedures for smoothing the data by 100 m should be described.**
    a. The 100 m smoothing is achieved by a boxcar average technique that is used in the 'smooth' function in the IDL programming language. The profiles are interpolated to 10 m vertical resolution and then the width of the smoothing window is specified to 10 points which equates to a 100 m average. This has been added to the text with the following statement:
       i. 'In this study, the MAGIC radiosonde refractivity profiles were first interpolated to a uniform 10 m vertical grid and then smoothed by a 100 m boxcar window to reduce the noise in the gradient profile resulting from the high sampling rate.'
    b. Side note that this is the same response as that for comment 14. Smoothing will be defined/described when first referred to at this point in the paper and then be referenced at the second point when smoothing of the bending is mentioned.

    **c.** The differences in smoothing window size will be stated explicitly in both references.

**11. L167: Explain what you mean by "the residue layer".**
    **a.** The use of "residue layer" was accidental and should have been "residual layer". It is used in reference to the location on the gradient profile below the PBL that showed the instance of a strong refractivity gradient that did not reach the critical value.
    **b.** This line was also noted by the second reviewer and has been deemed unnecessary in terms of the section and subsequently removed from the manuscript.

**12. Figure 2: When was this sounding taken? The figure caption for (a), (b), (c), (d) is out of sequence.**
    **a.** The sounding in this example was taken on 10/2/2013 at 0530 UTC.
        **i.** MAGIC: (Latitude, Longitude) (23.69˚N, -150.02˚E)
        **ii.** ERA5: (Latitude, Longitude) (23.75˚N, -150.00˚E)
    **b.** These profiles are also used in Figure 3. The caption has been rewritten to improve clarity as follows:
        i. Figure 2: Vertical profiles of refractivity ($N$ in deca-$N$-units, solid blue), temperature ($T$ in ˚C, dotted red) and specific humidity ($q$ in g kg$^{-1}$, dashed green) for (a) radiosonde at (23.69˚N, -150.02˚E) launched at 2012-10-02, 05:30 UTC, and (c) colocated ERA5 at (23.75˚N, -150.00˚E); and associated gradient profiles for radiosonde (b) and ERA5 (d). The horizontal dashed line highlights the height of the minimum gradient, i.e., PBLH. The paired horizontal dotted lines represent the bottom and top of the two ducting layers in the radiosonde profile (a and b) but only one in the ERA5 profile (c and d).

**13. L181: Add a reference to Abel integration.**
    **a.** Included 3 references to the end of this sentence:
        **i.** 1 previously listed in bibliography
            **1.** Sokolovskiy, 2001
        **ii.** Two references for specific Abel integration.
            **1.** Eshleman, V.R.: The radio occultation method for the study of planetary atmospheres, Planet. Space Sci., 21, 1521-1531, doi.org/10.1016/0032-0633(73)90059-7, 1973.
            **2.** Fjeldbo, G., and Eshleman, V.R.: The Atmosphere of Mars Analyzed by Integral Inversion of the Mariner IV Occultation Data, Planet. Space Sci., 16, 1035-1059, doi.org/10.1016/0032-0633(68)90020-2, 1968.

**14. L189: How did you perform 50 m vertical smoothing?**
   a. The 50 m smoothing is achieved by a boxcar average technique that is used in the 'smooth' function in the IDL programming language. The profiles are interpolated to 10 m vertical resolution and then the width of the smoothing window is specified to 5 points which equates to a 50 m average.
   b. More detail on smoothing method is added in response to comment 10.

**15. L247: What is the meaning of "moisture lapse"?**
   a. The phrase "moisture lapse" should have been "moisture lapse rate". The term refers to the decrease in moisture with height. This concept is analogous to the "temperature lapse rate", where a positive value indicating a decrease in temperature with increasing height.
   b. "Moisture lapse" was found in multiple places in the manuscript, and all have been changed to either "moisture lapse rate" or "sharp moisture lapse rate".

**16. L249: Cite Riehl (1979) or any other references at the end of this sentence.**
   a. Citation added:
      i. Riehl, H.: Climate and weather in the tropics. London: Academic Press. 611 pp. ISBN 0.12.588180.0.1979

**17. Section 3: It might be useful to plot a typical ERA5 sounding and compare it with the observed radiosonde sounding near Hawaii as well as off the California coast.**
   a. The authors considered adding a second radiosonde and ERA5 profile to represent conditions adjacent to the California coast as the initial manuscript was written. In the interest of space it was determined that the most sensible way to add a second set of profiles would be to introduce them in Figures 2 and 3. While this would add a visual of the difference at each end of the transect, the authors felt that the addition of overlaid profiles might create confusion within the illustrations used to describe the refractivity gradient, ducting layer, PBLH and $N$-bias. The authors further determined the 5°median statistics were sufficient in illustrating the behavior of the profiles by use of key variables PBLH, minimum refractivity gradient and sharpness in Figure 4 and $N$-bias profiles in Figures 7 and 8.

Reviewer References:
Riehl, H. 1979: Climate and weather in the tropics. London: Academic Press. 611 pp.

---

## Author Comment (AC2)

The authors thank the anonymous reviewer for their comments. Please see the individual responses to the reviewer's comments below.

**Reviewer #2 Comments – Major Revisions**

General comments:

Using high-resolution radiosondes from the MAGIC field campaign, the planetary boundary layer height (PBLH) along the transect between California and Hawaii are derived and discussed. In particular it is investigated how radio occultation data retrieval would underestimate the true refractivity below the PBLH, given perfect measurements (if radio occultation data had the very high vertical resolution of the radiosondes), when there is ducting. Results are compared to similar results obtained by using ERA5 data. In both cases (radiosondes and ERA5) data are forward-modelled and inverted in end-to-end simulations to obtain the underestimated refractivity profiles.

1. **I find the study interesting and worthy publication, but it is a bit unclear to me what is new and what has been found before. Has it been found before that the PBLH increases along the MAGIC transect from west to east? Or is this a new result? I suppose it is new to perform end-to-end simulations to study the ducting and PBLH variations along this transect and how it would underestimate radio occultation retrievals in this area. However, it is only in principle, because in practice the radio occultation vertical resolution is somewhat coarser than the resolution of the MAGIC radiosondes. I think this needs to be mentioned.**
   a. The authors appreciate the suggestions, and have modified the introduction to emphasize the unique contribution of this study, i.e., identification and illustration of the detailed characteristics of ducting, and assessment of its potential impact on GNSS RO through simulation study.
   b. Past studies have documented the PBLH in the MAGIC region from various data sources including the radiosonde and GNSS RO (e.g., Ho et al., 2015). The ducting induced $N$-bias in RO retrieval also has been recognized for a long time (Sokolovskiy, 2003). However, those studies did not provide detailed characteristics of ducting from radiosonde observations. Moreover, no comprehensive study on the impact of various ducting (e.g., height, strength, etc.) on GNSS RO retrieval has been conducted. Thus, the detailed analysis of the ducting characteristics, and end-to-end simulation of the MAGIC radiosondes are the new and major contribution of this study.
   c. The reviewer has a great point regarding the vertical resolution. The conventional belief is that GNSS RO has a vertical resolution of approximately 100 m in the lower troposphere. Gorbunov et al. (2004) demonstrated that radioholographic retrieval algorithms resolve atmospheric multipath problems, reduce RO biases in the moist lower troposphere, overcome the limitation from Fresnel diffraction, and improve the vertical resolution up to ~60 m. Although we did not specifically investigate into this issue, we did apply 100 m vertical smoothing on the radiosonde profiles, but not on ERA5 profiles, which has a resolution of less than 100 m below 1 km.

**d.** Added to beginning of the last paragraph in the introduction:

    i. "To comprehensively assess the potential impact of ducting on GNSS RO retrievals, we begin by constructing a detailed ground truth of PBL ducting statistics. This is derived from an extensive set of high-resolution radiosonde data over the northeastern Pacific Ocean, a region known for prevailing ducting conditions. Subsequently, we conduct a simulation study using the radiosonde data to evaluate the *N*-biases caused by varying ducting characteristics."

    ii. In Section 2.2, "In this study, the MAGIC radiosonde refractivity profiles were first interpolated to a uniform 10 m vertical grid and then smoothed by a 100 m boxcar window to reduce the noise in the gradient profile resulting from the high sampling rate. *Moreover, the 100 m smoothed radiosonde will be more consistent with the vertical resolution of GNSS RO measurements (e.g., Gorbunov et al., 2004).*"

**e.** References:

    i. Gorbunov, M. E., H. H. Benzon, A. S. Jensen, M. S. Lohmann, and A. S. Nielsen, 2004: Comparative analysis of radio occultation processing approaches based on Fourier integral operators. Radio Sci., 39, RS6004, https://doi.org/10.1029/2003RS002916.

    ii. Ho, S.-P., L. Peng, R. A. Anthes, Y.-H. Kuo, and H.-C. Lin, 2015: Marine boundary layer heights and their longitudinal, diurnal and inter-seasonal variability in the southeast Pacific using COSMIC, CALIOP, and radiosonde data. J. Climate, 28, 2856–2872, https://doi.org/10.1175/JCLI-D-14-00238.1.

    iii. Sokolovskiy, S. V.: Effect of super refraction on inversions of radio occultation signals in the lower troposphere. Radio Sci., 38 (3), https://doi.org/10.1029/2002RS002728, 2003.

2. **For the same reason as above, and because the study does not actually present radio occultation data, I suggest to insert 'Potential' in front of 'Impact' in the title.**

    **a.** Thank you for this suggestion. The authors have agreed, and the title has been updated:

        **i.** 'Assessing the Ducting Phenomenon and its Potential Impact on GNSS Radio Occultation Refractivity Retrievals over the Northeast Pacific Ocean using Radiosondes and Global Reanalysis'

3. **Throughout the manuscript the authors refer to their results being a 'climatology'. I think that is a misuse of the term 'climatology'. There is only one year of data which is averaged in longitude bins along the transect without taking seasonal variations into account. I wouldn't consider that a climatology. I suggest not to call it 'ducting climatology', or 'N-bias climatology', etc. Most of the results are statistics based on that one year of data, so in most places 'climatology' could be replaced by 'statistics', or just removed.**

    **a.** Thank you for this comment. The authors have replaced 'climatology' with 'statistics'.

4. **I think the paper could be shortened by taking out some paragraphs (see specifics below), and perhaps even by taking out all of section 3.3.3 that describes some rather obvious correlations. They seem to be of little importance. In my opinion, the paper would be better (and the rest of the study is sufficient) without section 3.3.3.**
   a. The authors believe that Sect. 3.3.3 includes some important information and have tried to make the main points clearer. Firstly, the ducting characteristics change quite significantly along the transect, especially the ducting strength and ducting height. Figures 10 & 11 demonstrate the consistent linear relationship between the PBLH and ducting height as well as the linear relationship between the ducting characteristics and $N$-biases (Fig. 11). The relationship is relatively independent of the spatial location of the profiles. Further, the difference between radiosonde and ERA5 observations shows the potential impact of the uneven vertical sampling in ERA5.
   b. This answer will also address the similar question stated in 'Specific comments' point 7 below.

5. **I find that parts of the manuscript are badly written, and there are several mistakes in the figure captions (see specifics below). In some parts of the manuscript, almost every sentence needs revision.**
   a. The authors have made significant effort to improve the overall writing of the manuscript. It is our sincere hope that the reviewer recognizes these efforts and deems the result of such detailed comments as a worthy improvement.

Specific comments:

1. **lines 137-139: It is not clear how the root mean square error of the refractivity gradient profile is calculated. Is it really an 'error'? Compared to what? Over which vertical interval? Please clarify. Please also write in the text what each of the symbols in eq. 2 stand for. Why are you writing X when it is N (I think)?**
   a. The variables '$X$' used in the original source (Ao et al., 2012) cited in the paper is replaced with refractivity ($N$) for clarification.
   b. The "error" in sharpness parameter definition has been removed, and the manuscript has been updated as follows:
      i. 'To assess robust PBLH detection with gradient method, Ao et al. (2012) introduced the sharpness parameter ($\widetilde{N}'$) to measure the relative magnitude of the minimum gradient, which is defined as the ratio of the minimum vertical refractivity gradient ($N'_{min}$) to the root mean square ($N'_{RMS}$) of the refractivity gradient profile from surface to 5 km as follows:'

2. **Fig. 2: Please provide information about latitude, date, and time, for this example.**
   a. Text added to main body and caption to reflect the date and time (2012-10-02 at 05:30 UTC) and location of the radiosonde (23.69˚N, −150.02˚E) and ERA5 (23.75˚N, −150.00˚E)

3. **Text before Fig. 2: Please provide information on how the gradients in Fig. 2 were calculated. In the ERA5 refractivity gradient there are sawtooth-like features just above the minimum gradient (and elsewhere in the profile to a lesser degree), and levels seem very unevenly spaced. Are these features real (in the model) or an artifact of how the gradients are calculated?**

   a. Thanks for catching this. Both the radiosonde and ERA5 data were vertically interpolated to a 10 m resolution. The refractivity gradient for both profiles is calculated by taking the derivative of the interpolated profile with respect to height. In this case, the derivative is calculated with a three point Lagrangian interpolation technique that is part of the IDL 'deriv' function. Note that while the saw-toothed feature mentioned is due to the vertical derivative being calculated from the interpolated profile, this is only noticed in the plotting and does NOT impact the result of the study.

4. **line 241-243: "This results in a sharper refractivity gradient caused by the frequent residual layer (below 1 km) as compared to the actual PBLH near 2 km.". How do you know which one is the 'actual PBLH' when there are such residual layers with apparently sharper gradients? What do you consider to be the definition of the PBL in this study? Please discuss this in the text.**

   a. The definition of the PBL height in this study is simply defined as the height of the minimum refractivity gradient as is defined within the text of lines 124-129. The detailed description of the gradient method for PBLH detection is added in Section 2.2. However, the physical meaning/definition of PBLH can vary (e.g., Seidel et al., 2010).

   b. Over the western segment of the transect (near Hawaii), there are often two gradient layers (one at ~1km and another at ~2km) with comparably $N$-gradients (e.g., Fig. 2). Generally, the higher-level gradient layer exhibits a slightly larger $N$-gradient and will be identified as the PBLH based on the gradient method. Such a PBLH around 2 km is widely known as the PBL trade-wind inversion. However, the lower-level gradient layer around 1 km, is a result of the ERA5 having more sampling layers below 1 km (~19 model levels) than higher levels (~8 levels from 1-2 km). The gradient method could identify the lower-level gradient layer as the PBLH due to the artificially higher sampling rate at lower levels. Such discrepancy in PBLH between ERA5 and radiosonde is seen in Fig. 4a. (continued on next page)

**c.** Text of this line has been changed to the following.

  i. "Such a discrepancy could be due to the sensitivity of gradient method to the vertical resolution of the data. Over the western segment of the transect (near Hawaii), two major gradient layers (one at ~1 km and the other at ~2 km) with comparable refractivity gradients are often observed (e.g., Fig. 2). The gradient layer at around 2 km is well-known as the trade-wind inversion. While the lower-level gradient layer at ~1 km, is generally called the mixing layer. Note the radiosonde data exhibit consistent vertical sampling (~125 points per km) below ~3 km and resolve both layers well. However, the ERA5 data have uneven vertical sampling intervals that increase with height with $10 - 100$ m resolution below 1 km, $100 - 160$ m within 1-2 km, and $160 - 200$ m within 2-3 km. Therefore, the ERA5 data are more likely to resolve the sharp gradient structure below 1 km than the one at higher altitude. This could result in resolving the mixing layer (below 1 km) as the sharpest refractivity gradient, instead of the trade-wind inversion near 2 km in the ERA5 data."

**5. Fig. 5b: How did you calculate such narrow ducting thicknesses for ERA5, in particular in the western part of the transect? The median thicknesses are between 50 and 100 m in the western part, while the ducting height is within 1-2 km where there are only 8 levels in ERA5 (noted in line 115). Please provide more detailed information on the calculation of the ducting thickness.**

  **a.** As described in Section 2.1, the raw vertical resolution of radiosonde is ~ 8 m below 3 km, whereas the ERA5 data have $10 - 100$ m resolution below 1 km, $100 - 160$ m within 1-2 km, and $160 - 200$ m within 2-3 km.  All the radiosonde and ERA5 $N$-profiles were then interpolated into 10 m vertical grids. Therefore, both the radiosonde and ERA5 data can resolve the sub-100 m ducting layer (seen in Fig. 6). Near the western boundary of the transect the ERA5 data tend to identify the shallow mixing layer below 1 km as the dominant ducting layer instead of the trade-inversion above (as discussed in #4 above).

**6. line 297: Why 'median' here? It is individual cases in Fig. 6, right?**

  **a.** Thank you for catching this error. The word 'median' has been removed from this sentence.

**7. line 320-326: I think the discussion about the small difference between the PBLH and the maximum N-bias is a bit academic. The exact size probably depends on the particular method of calculating the PBLH and the end-to-end simulations, including the smoothing that is involved. Is it important? The differences are seen later in Fig. 8, which seems sufficient.**

  **a.** The authors feel the difference between the PBL height and maximum $N$-bias height warrants discussion. But we agree that the paragraph should be moved to Section 3.3.2 and merged with the discussion of the Fig. 8.

8. **line 327-331: It seems that this is discussing results shown later in Fig. 8. I suggest to move this text to section 3.3.2.**
   a. The authors agree with the reviewer. This paragraph was revised and merged with the rest of the Figure 8 discussion.
   b. See Technical corrections 19 (a and b) for description of changes.

9. **Table 1: The median numbers here are a bit off from the numbers discussed in the text (end of section 3.3.1). Please revise either the text or the numbers in the table.**
   a. Thank you catching this. The table figures were not updated, but the in-text figures were correct. The Table has now been updated.

10. **line 348-349: I suppose you are discussing the ERA5 data here, but it should be made more clear.**
    a. Thanks for the comment. "ERA5" has been added to the sentence.

11. **line 374-379: I didn't quite understand this paragraph. For example, I don't understand that a "much higher ducting height and larger variation leads to smoothed and much smaller median N-gradient values". Why are you using the word 'leads'? Does the former cause the latter? I think you are trying to say that without the normalization the N-bias would be smaller than it is with normalization, but it is not clear. Please clarify and revise this paragraph.**
    a. Yes, the reviewer's understanding is correct. We have updated the paragraph for better clarification as below:
       i. 'Note that normalizing each $N$-bias profile to the PBLH preserves the magnitude of the $N$-bias with various heights. Therefore, the relatively large normalized $N$-bias observed near Hawaii indicates more persistent ducting over the trade-cumulus boundary layer regime compared to the transition region in the middle of the transect at -147.5˚E (Fig. 8a).'

12. **line 422-426: I don't understand the sentence: "... it is interesting to note that the difference in the correlation of the radiosonde (−0.83) and the ERA5 (−0.84) does not lie in the observations with the larger magnitude peak N- bias, but in those closer to zero as the radiosonde data clearly centers below the regression line and trends above while the ERA5 with peak N-bias less than 5% are centered around the regression line.". Are you talking about the very small difference between 0.83 and 0.84? I can't see how you can conclude that this difference comes from the data with small maximum N-bias. In any case I don't think it is important. Is the sentence necessary? Please clarify if it is.**
    a. We agree with the reviewer and removed the sentence.

**13. line 434: I suggest to replace 'climatology and the impact of' with 'and the'. The study did not investigate the ducting climatology since there was only one year of data. The study did not investigate the impact of the biases (impact on what?). The sentence also needs to clarify that it is in relation to radio occultation retrievals that there would be biases. Please revise.**

    **a.** Thank you for the comment. The sentence has been modified to improve the clarity as follows:

        **i.** 'In this study, radiosonde profiles from the MAGIC field campaign have been analyzed to investigate ducting characteristics and the induced systematic refractivity biases in GNSS RO retrievals.'

**14. line 438: I don't understand 'at a well-defined PBL throughout the transect' in this sentence? Could it be removed?**

    **a.** The sentence has been removed.

**15. line 458-459: I don't think this is correct: "While this segment of the transect also coincides with a better sampling rate for the ERA5 data (~40 m vertical resolution)". Isn't the resolution of ERA5 the same throughout the transect? Maybe you mean that because the PBLH is lower in the eastern part, the ERA5 vertical resolution around the height of the ducting layer is higher in the eastern part, but it is not clear. Please clarify.**

    **a.** You are correct. The sampling rate of the ERA5 is the same throughout the transect.

    **b.** The statement was meant to refer to the fact that since the refractivity gradient is stronger and PBL is at a lower altitude, the higher vertical resolution of the ERA5 is more likely to identify the PBL at a height similar to that which is identified by the radiosonde.

    **c.** This line has been revised to improve the clarity of the statement, see comment 16.c.i. below.

**16. line 462-464: I don't think you can conclude that the differences that you see between the radiosondes and ERA5 are due to the 'limited number of model levels in ERA5 near 2 km'. There is no investigation of the impact of the lower resolution in this study. In principle, ERA5 could be underestimating the heights for other reasons. Please be more moderate in the conclusions.**

    **a.** This is a fair point. Generally, the authors chose to include this line as a reference to the large discrepancy of the PBL height on the western side of the analysis transect where the difference between the median PBL height between MAGIC and ERA5 exceeds 800 m. The authors felt this was likely due to the number of ERA5 data points between 1 and 2 km was an average of 8 where as the number of radiosonde data points for each 1 km layer from the surface to 3 km are an average of 8 m (125 observations per km). The radiosonde data are more likely to observe the true location of the minimum gradient height and thereby the PBLH. Additionally, the reduced sharpness of the gradient in the western portion of the transect mean the minimum gradient is not as well defined and, in turn, height of the PBLH identified by the minimum gradient could be washed out due to the natural smoothness of the ERA5 profile.

    **b.** The sentence is updated as follows:

        **i.** 'It is worth noting that the PBL over the western portion of the transect near Hawaii frequently shows two major gradient layers (a mixing layer at ~1 km and the trade-inversion at ~2 km), with comparable refractivity gradients (e.g., Fig. 2). The much lower PBLH seen in ERA5 in this region is likely due, in part, to the decreasing number of model levels in ERA5 at higher altitude, which could lead to higher possibility of identifying the lower gradient layer as the PBLH. However, the impact of the vertical resolution on the performance of gradient method for PBLH detection has not been performed in this study and warrants more comprehensive study in the future.'

**17. line 465-468: I think this 'future study' paragraph should be removed. It does not belong in a conclusions section, and there is no need for it.**

    **a.** The authors agree and the 'future study' paragraph has been removed.

**Technical corrections:**

**1. line 21-22: I think either the 'and' in line 21 should be replaced with a comma, or the ';' in line 22 should be a comma. Maybe correlation should be plural. Please revise sentence.**

    **a.** The last line has been changed:

        **i.** 'Further, the underestimation of the $N$-bias in the ERA5 data increases in magnitude westward, the correlations between the $N$-bias with the minimum gradient and sharpness all remaining strong.'

**2. line 32-36: I don't think you need 'etc' in line 36 when you have 'such as' in line 32.**

    **a.** Thanks, the "etc." has been removed.

3. **line 111-112: Maybe it should be 'reanalysis', not 'Reanalysis' in line 111. I think there is no need for 'reanalysis' in line 112, as it is already part of the ERA5 acronym.**
    a. The word reanalysis has been removed as it is referenced in the description of the acronym.

4. **line 126: I don't understand "the minimum refractivity describes the largest magnitude value." Please revise the sentence.**
    a. The sentence has been removed.

5. **line 159 and Fig. 2: I think it should be 10 x N-units (not 1/10). Like with m and km, if you plot something as a function of height/1000, where height is in m, the axis unit becomes km (1000 x m).**
    a. The reviewer's point is understood. In this case, the refractivity values are an order of magnitude larger than those of temperature, and mixing ratio. As a result, the $N$ value must be divided by 10 in order to fit on the x-axis with the units used for temperature (˚C) and specific humidity (g kg$^{-1}$).

6. **Fig. 2: I suppose T is in degree Celsius here (not kelvin).**
    a. The unit has been corrected in the caption and in the Figure 2 x-axis title.

7. **line 167-168: I think it should be 'a residual layer' instead of 'the residual layer'. There has been no mention of this layer earlier in the text. Something is not right with line 168, maybe an 'and' is missing. Please revise.**
    a. The other reviewer also brought both points up. The line has been revised:
        i. 'The PBLH of the radiosonde (2.10 km) is almost identical to the colocated ERA5 (2.14 km) and the "dominant" ducting layer near the PBLH demonstrates similar thickness. However, a second, weaker ducting layer seen in the radiosonde above the PBLH was not captured by the ERA5.'

8. **line 180: '1-dimensional' instead of '1-dimentional'**
    a. Changed to "1-dimensional".

9. **line 184: I think it should be 'increases' instead of 'decreases' (if it is 'with height' as written).**
    a. The reviewer is correct; "decreases" has been changed to "increases".

10. **line 194-195: The word 'respectively' is used here to describe what is in Fig. 3a and 3e, but it is used wrongly. What is seen in the two figures are refractivity profiles from the radiosonde and the ERA5 data, respectively. It is not the input refractivity profile and corresponding Abel refractivity retrieval, respectively. Please revise.**
    a. The sentence has been revised as follows:
        i. 'Figures 3a and 3e show refractivity profiles from the radiosonde ($N_{rds}$) and the colocated ERA5 ($N_{ERA5}$) data, as well as their corresponding Abel refractivity retrievals ($N_{Abel}$).'

**11. Fig 3 caption: I think "10 km" should be "4 km" and "minimum gradient" should be "refractivity gradient". The last sentence could be revised to be more precise, for example: "The same is shown in panels e-h for the co-located ERA5 profile".**

    **a.** The authors agree with the reviewer and all three suggested changes have been made.

    **b.** Revised caption for Fig. 3:

        i. 'Figure 3: End-to-end simulation data for MAGIC radiosonde launched at 0530 UTC on 20121002 showing: (a) $N_{Obs}$ (solid red) and $N_{Abel}$ (blue dashed) from surface to 4 km; (b) PBLH adjusted $N$-bias (($N_{Abel} - N_{Obs}$)/$N_{Obs}$ x 100); (c) refractivity gradient and (d) bending angle vs. impact parameter. The same is shown in panels e-h for the colocated ERA5 profile.'

**12. line 211: "Out of a total of 583 ..., quality control has been implemented ...". I think I understand what you want to say, but literally it makes little sense. Please revise the sentence.**

    **a.** First sentence of the paragraph was changed:

        **i.** 'Quality control for radiosonde (and co-located ERA5) profiles was based on five key criteria.'

**13. Fig. 4 caption: I believe b) and c) should be interchanged (also in lines 232-233). It seems that the MAD error bars are dotted for both radiosondes and ERA5, whereas it is the lines connecting the points that are dashed or dot-dashed. Please revise.**

    **a.** The authors appreciate the reviewer for bringing their attention to this error and the line styles of median and MAD were switched.

        **i.** Text in caption and body have been changed to accurately reflect line order as:

            **1.** '…value of PBLH (a), minimum gradient (b) and sharpness (c) along the transect.'

        **ii.** Text in caption has been changed to accurately reflect line texture as:

            **1.** '…for MAGIC (median in red circle and dashed line, MAD in red dotted error bars) and ERA5 (median in blue diamond and dot-dashed line, MAD in blue dotted error bars)'

**14. Reference to figures: Often references to figures are made in parentheses, in particular for Fig. 5 in Section 3.2, but also elsewhere. Probably references to figures should be made in text (without parentheses) at least the first time around (I am not sure what the AMT guidelines say).**

    **a.** The AMT guidelines for figure references do not specify the use of in text vs. parentheses for the first use, only that in text should be abbreviated "Fig." when used in running text unless it comes at the beginning of a sentence in which case "Figure" should be used.

    **b.** Time was taken to scan for the first reference to each figure and ensure that it was made in text instead of in parentheses while following the aforementioned AMT guidelines.

**15. Fig. 5b: This panel has a different x-axis coverage than the other tree panels. Please adjust.**

    **a.** Figure 5b has been adjusted so the x-axis coverage is uniform for all four windows in the plot.

**16. Fig. 5 caption: I suppose it should not be 'error bars' in '(median in blue diamond and dot-dashed error bars)'. Please revise.**

    a. Yes. Caption section has been changed to median in blue diamond and dot-dashed line, MAD in blue-dotted error bar).

**17. line 315: Should the 'e.g.' be 'i.e.'?**

    **a.** Yes. 'e.g.' has been changed to 'i.e.'.

**18. line 318: Should 'between' be 'of'?**

    **a.** Yes. 'Between' has been changed to 'of'.

**19. line 327-331: I don't understand what is meant by 'favors the radiosonde data' here. Could it be written differently? There are ending parentheses without beginning parentheses in this paragraph. Please revise.**

    **a.** The sentence has been modified and moved to Section 3.3.2 for Fig. 8 discussion:

        **i.** 'The maximum peak $N$-bias ($-7.86\%$) in the radiosonde data is located at the easternmost of the transect near California ($-122.5°E$). Whereas the minimum peak $N$-bias ($-4.37\%$) is located near the center of the transect ($-147.5°E$). Similarly, the ERA5 also show the maximum peak $N$-bias ($-5.92\%$) near California ($-122.5°E$). However, the minimum peak $N$-bias ($-0.77\%$) is found near Hawaii ($-157.5°$). Overall, the $N$-bias in ERA5 are smaller than radiosonde in all bins.'

**20. line 344-346: This sentence does not make sense to me: "The radiosonde N-bias variation shows a minimum magnitude of near the center of the transect and two of the largest magnitude difference values of as the bookends while the ERA5 N-bias values have a larger range but peak values ($-5.41\%$ to $-6.23\%$) in the three bins closest to California". Could it be written differently?**

    **a.** The paragraph has been rewritten for clarity as follows:

        **i.** 'However, a noticeable difference exists between the ERA5 and radiosonde profiles for the two westernmost longitude bins ($-157.5°E$ and $-152.5°E$) where the ERA5 reveals a much lower and weaker $N$-bias than the MAGIC data.'

**21. line 440: I suggest to use 'California', 'Hawaii' and 'refractivity' throughout the abstract instead of 'CA', 'HI', and 'N'.**

    **a.** Authors agree with the full name replacement of abbreviations for California and Hawaii.

    **b.** The use of $N$ in reference to refractivity mainly used when referring to the refractivity gradient ($N$-gradient) and bias ($N$-bias).

        **i.** All instances of '$N$-gradient' have been changed to 'refractivity gradient' or just 'gradient' when refractivity is already referenced within the sentence.

        **ii.** The authors planned to reference the bias within this paper and, as such, defined '$N$-bias' as an abbreviation for refractivity bias. Since this is the case, the authors believe that keeping the reference to '$N$-bias' is acceptable and should remain throughout the paper.

**22. line 450-453: Correlation between the PBLH and the height of the maximum N-bias is mentioned twice. Please revise.**

    **a.** Removed second mention of the correlation.

**23. line 454: Past tense is used here, whereas the next sentence is in present tense. Please be consistent.**

    **a.** Noted. Changed to present tense for consistency with the following sentence.

**24. line 455: I suggest to say 'opposite' instead of 'reverse'.**

    **a.** 'reverse' has changed to 'opposite'.

**25. line 538: doi.org/10.1175/HTECH-D-19-0206.1 is wrong. It should be 'JTECH'.**

    **a.** Changed to 'JTECH'

---

## Author Response (AR2)

Reviewer #2 second round of comments – Major revisions

**Report #1**

Submitted on 28 Jan 2024
Anonymous referee #2

**Anonymous during peer-review: Yes** No
**Anonymous in acknowledgements of published article: Yes** No

**Checklist for reviewers**

| | |
|---|---|
| **1) Scientific significance**
Does the manuscript represent a substantial contribution to scientific progress within the scope of this journal (substantial new concepts, ideas, methods, or data)? | Excellent **Good** Fair Poor |
| **2) Scientific quality**
Are the scientific approaches and applied methods valid? Are the results discussed in an appropriate and balanced way (consideration of related work, including appropriate references)? Note that papers do not necessarily need to be long to be scientifically sound. | Excellent Good **Fair** Poor |
| **3) Presentation quality**
Are the scientific results and conclusions presented in a clear, concise, and well structured way (number and quality of figures/tables, appropriate use of English language)? | Excellent Good **Fair** Poor |

**For final publication, the manuscript should be**

accepted as is

accepted subject to technical corrections

accepted subject to minor revisions

**reconsidered after major revisions**

rejected

**Were a revised manuscript to be sent for another round of reviews:**
**I would be willing to review the revised manuscript.**
I would not be willing to review the revised manuscript.

Thank you to the reviewers for constructive comments. Our responses to the reviewer comments are below.

**Responses to Reviewer #2:**
I'm happy with most answers to my questions and general improvements to the manuscript. However, there are still three major issues (numbered below) that needs to be addressed before I can recommend publication.

1. **The sawtooth-like features in Fig. 2d are not adequately explained. In their answer to my question (number 3 under specific comments), the authors say that these features do not impact the result of the study. I am not convinced, and I don't understand from the author's explanation how they appear. If they are an artefact of how the data are interpolated or derivatives are calculated (as indicated in the answer), then I think the method needs to be changed to one that does not introduce such artefacts. Gradients are an essential part of this paper and artificial gradients shouldn't be accidentally introduced by the method. In any case, please explain in the text how the data are interpolated to the 10 m vertical grid (is it linear interpolation or something else?) and how the derivatives are calculated. If the authors insist that the current method is sound, then they need to mention the reason for the sawtooth-like features in the text and explain why they do not impact the results.**

- For this study, the only gradient of importance is the absolute minimum gradient of the profile. In the specific example of Figure 2d, there are two noticeable gradients (near 0.5 km and 2.1 km), the latter of which has a value of $-200$ N-units km$^{-1}$. Both are a result of sharp moisture gradients. The sawtooth features above the minimum gradient are nearly four times smaller than the value that is used to define the PBLH; further, if they do not qualify as ducting and therefore if the moisture induced refractivity gradients were absent, the profile would not be used in the study.
- The non-linear vertical grid in ERA5 can create artificial steps in the refractivity profile at very fine scales. It is unwise to perform too much smoothing to remove the features, since it would low-bias the PBLH/ducting height in addition to resulting in the loss of a significant number of cases where the refractivity gradient exceeds the ducting threshold.
- The quality control process that is outlined at the beginning of Section 3 defines the minimum gradient as the largest magnitude negative gradient of the profile; further, the minimum gradient threshold value is $-157$ N-units km$^{-1}$. It can be seen in Figure 2c and 2d that any minor value that is introduced as a result of the calculation of the derivative does not impact the location or presence of the absolute minimum gradient in any way.
- The 'deriv' function part of the IDL library (https://www.nv5geospatialsoftware.com/docs/DERIV.html) and "uses a three-point (quadratic) Lagrangian interpolation to compute the derivative of an evenly-spaced or unevenly-spaced array of data."
- The interpolation is performed at the beginning of the analysis and is a quadratic interpolation that filters NaN values so as not to create data from an undefined point. The first step identifies the beginning and the end of the input height array so values are not extrapolated beyond the value of the original domain. Second, the original refractivity and height profiles are used as input values and the uniform height profile (surface to 100

km with resolution of 10 m) is used to interpolate the refractivity profile to the same resolution.

- The following text will be added to the end of Section 2.2 where the interpolation of both data sets is discussed:
  - "In the case of both data sets, quadratic interpolation is used to translate the refractivity profiles from their raw height value to a uniform height profile which is necessary for a sound statistical comparison."
- The following text detailing the gradient calculation will be added to the beginning of Section 2.3:
  - "The refractivity gradient profile is calculated by differentiating the 10 m interpolated refractivity profile with respect to height. The height array is an evenly spaced array from the surface to 100 km with a vertical resolution of 10 m, it is the array that is used for the interpolation process."
- The following text acknowledging the "sawtooth-like" structure above the minimum gradient will be added to the end of Section 2.3:
  - "Note that the weak gradients seen above the minimum in the ERA5 refractivity gradient (Fig. 2d) are a result of the vertical derivative being calculated from the interpolated ERA5 refractivity profile and do not appear for larger interpolation intervals suggesting that the non-linearity of the ERA5 vertical grid at this height affects the vertical gradient. These features of approximately 15 N-units km$^{-1}$ magnitude are only noticed in the plotting and do not impact the results of the study, as only the moisture-induced minimum gradient values are large enough in magnitude to exceed the minimum gradient threshold."

2. **I still do not understand how it is possible to get such narrow ducting thickness for ERA5. In their answer to my question (number 5 under specific comments), the authors explain that it is because the data are interpolated to a 10 m vertical grid. However, that should not increase the underlying resolution. In particular I am concerned when I see the gradient profile in Fig. 2d with the sawtooth-like features that appear to be artificial (due to the vertical derivative being calculated from the interpolated profile). Couldn't these features result in a wrong and too narrow ducting thickness? Could they also influence the results in Fig. 6?**

- The narrow ducting thickness can occur as a result of a gradient value that narrowly exceeds the ducting threshold of −157 N-units km$^{-1}$. In the case of such an occurrence, it is plausible that the resulting ducting layer could be interpreted with a thickness of only 10 m as the quadratic interpolation could limit a 10 m vertical resolution profile to one point. The authors justify this as acceptable because the interpolated value is providing a more realistic ducting layer thickness. As an example, an ERA5 profile that is not interpolated can have a minimum gradient value of −160 N-units km$^{-1}$ and if the raw profile resolution at that point is 50 m, one could argue the ducting layer thickness is 50 m. However, a 50 m ducting layer thickness could be just as artificial as the reviewer is claiming for a 10 m thickness. In this example the 10 m interpolation is a more accurate assessment of the ducting layer because it allows for estimation based on a uniform height profile and not for an assumption of thickness to be made solely by the non-linear model layers.
- The "sawtooth–like features" have been discussed at length in point 1 above.

3. **I still think the paper would be better without section 3.3.3. But if it is kept, I have the following questions and recommendations: a) How come the linear regression line in Fig. 10a has a slope larger than 1.0 and that most points are above the line in the lower part given that there are a number of outliers well above the diagonal? This seems counter-intuitive. Please check that it is correct. b) I suppose the linear regressions in Figs. 10 and 11 are done with the variables on the x-axes as independent variables and those on the y-axes as dependent variables. So in Fig. 10 it is the PBL height, and in Fig. 11 it is the peak N-bias that is considered as the independent variable. However, in this context, PBL height and peak N-bias are not actual independent variables. Likewise, height if peak N-bias, minimum gradient, and sharpness parameter are not dependent variables as a function of some independent variable. To me, linear regression makes little sense in this case. The reason that it makes little sense is that if you were to interchange the axes, thereby treating the other three variables as independent, you would get different regression lines. I'm not referring to the mirroring in the diagonal. The lines become profoundly different with different slopes and intercepts (see e.g., https://www.reddit.com/r/statistics/comments/12s21os/q_why_doesnt_flipping_the_axes_of_a_scatterplot/). It may not be by much (it depends on the scatter), but enough to question the purpose of the linear regression when you don't have an actual independent variable. Please consider not including the regression lines in these figures.**
- Section 3.3.3 has been removed.

Besides these issues, there are still many mistakes in the text. Some of them I did not catch in my first review, others have appeared in new sentences. I list them all below.

- line 58: **One ";" too many (between Ao et al. and Guo et al.). Commas are missing before the years.**
  - This was changed in response to the reviewer's comment on line 187 that the in-text citation order was not chronological.

- line 115: **I think it should be 'the' refractivity field. But I'm not sure the sentence makes sense using 'which'. The sentence in the original manuscript makes better sense to me, but here parts that before made sense has been left out.**
  - The last sentence has been restored to the original version: "Finally, ducting is prevalent throughout the domain over which the observations were captured creating an opportunity to perform an analysis over a natural cross-section of X (zonal) and Z (vertical) dimensions."

- lines 119-120: **In their answer to me comment on the use of the word 'reanalysis' here (number 3 under technical corrections), the authors write that "the word reanalysis has been removed ...". However, it doesn't seem to have been implemented in the revised manuscript. Please revise.**
  - The word 'reanalysis' has been removed, apologies for the redundancy.

- line 138: **'Seidal' should be 'Seidel'.**
  - Thank you for catching this error, the authors meant no disrespect to Dr. Seidel by the misspelling, and the change has been made.

- line 139: **A 'the' is missing before 'gradient method'.**
  - This line has been removed, please see response to line 140 below.

- line 140: **It seems that the definitions of N'_min and N'_RMS got lost. It is in the answer to one of my questions in the first review (number 1 under specific comments).**
  - The omission of the definitions for $N'_{min}$ and $N'_{RMS}$ was unintentional. However, since the authors have decided to take the reviewer's advice and remove Section 3.3.3 from the manuscript, discussion of the variables is no longer needed.
  - The sentence beginning on line 142 "To assess the robustness…" and continuing to line 145 "… to 5 km as follows:" as well as Eq. (2) on line 146 have been removed.

- line 142: **I am not sure what it means that "Each refractivity gradient profile can then be filtered...". I think you mean that one could filter out (remove from the study) those profiles where the sharpness parameter is not large enough, but that you did not do that in this study. Is that right?**
  - The reviewer is correct, that was the intended message with that sentence. The reviewer is also correct that 'filter' was not used in this study. Since this is the case, the line "Each refractivity gradient profile can then be filtered to identify the PBLH values with sharpness parameter exceeding a specific threshold, thus increasing the robustness of PBLH detection" has been removed from the paragraph.

- line 151: **Remove 'near surface' in this sentence.**
  - The text 'near surface' has been removed.

- line 164: **Would it be better to say more directly: "The ducting layer height is defined as the height of the top of the ducting layer"?**
  - Yes, that seems to be a much simpler way to state the definition, the text has been changed.

- line 166 and Fig. 2: **I think that 'N-units x 1/10' is still the incorrect unit. The authors answer to my comment on this (number 5 under technical corrections) is understood, but a correction does not seem to have been implemented in the revised manuscript. In their answer 12 to reviewer #1, the authors use deca N-units (deca means 10). I think that would be more correct. Alternatively, one could add numbers indicating the N-units on a second x-axis on top of the panels in question.**
  - Thank you for bringing the point to our attention. When the initial changes were made, the '1/10 x N in N-units' was unintentionally left incorrect as it was not realized that '1/10 x N in N-units' was contradictory.
  - Referring to the previous comment (number 5 in reviewer #2's technical correction), the author's do not feel a second axis is necessary but the units can be clarified by listing refractivity units as '$N$-units/10' for this specific figure since the refractivity values are on the order of $10^2$ below 3 km altitude.

o   References for refractivity in figure 2 (main text and caption) have been changed to '$N$-units/10'

- line 187: **Comma should be semicolon between 1973 and Fjeldbo. Is it a new rule in AMT to put the latest publication first in lists of citations? Usually it is the other way around (oldest first).**
    o   Comma was replaced with a semicolon.
    o   In-text citations have been reordered chronologically.

- line 201: **Perhaps (N_rds) should be (N_MAGIC) to be consistent with Fig. 3.**
    o   Agreed. $N_{rds}$ has been changed to $N_{MAGIC}$.

- line 204: **Perhaps skip (N_Abel - N_Obs)/N_obs here and in the Fig. 3 caption. It is not explained what N_Obs is, but I think things could be said without the use of symbols here. The Fig. 3 caption would have to be modified a bit.**
    o   $((N_{Abel}-N_{Obs})/N_{Obs})$ has been removed from line 206.
    o   Caption for Figure 3 has been modified to the following"
        - "End-to-end simulation results for MAGIC radiosonde launched at 0530 UTC on 20121002 showing: (a) $N_{MAGIC}$ (solid red) and $N_{Abel}$ (blue dashed) from surface to 4 km; (b) PBLH adjusted $N$-bias; (c) refractivity gradient and (d) bending angle vs. impact parameter. Panels e-h show end-to-end simulation results for the colocated ERA5 profile."

- lines 200-204: **These lines could be more complete in telling what is in Figure 3, for example something like: 'Figures 3a and 3e show refractivity profiles from the radiosonde (N_MAGIC) and the collocated ERA5 (N_ERA5) data as well as their corresponding Abel refractivity retrievals (N_Abel). The refractivity gradients are shown in Figures 3c and 3g. The PBLH is marked by a horizontal dotted line. The peak bending angles in Figures 3d and 3h are consistent with the sharp refractivity gradients (impact height typically being a few km larger than the height). Figure 3b shows the fractional N-bias (in percent) between the simulated Abel retrieved RO refractivity profile and the radiosonde, whereas Figure 3f shows the same for the ERA5 profile.'**
    o   The reviewer's rearrangement of these lines results in a better explanation of the information presented in Figure 3. The original text has been changed to the following:
        - "Figures 3a and 3e show refractivity profiles from the radiosonde ($N_{MAGIC}$) and the colocated ERA5 ($N_{ERA5}$) data as well as their corresponding Abel refractivity retrievals ($N_{Abel}$). The refractivity gradients are shown in Figures 3c and 3g. The derived PBLH is marked by a horizontal dotted line. The peak bending angles in Figures 3d and 3h are consistent with the sharp refractivity gradient. Figure 3b shows the fractional $N$-bias (in percent) between the simulated Abel retrieved RO refractivity profile and the radiosonde, whereas Figure 3f shows the same for the ERA5 profile."

- line 206 (and 321): **Is 'normalized' really the right word to use here? Would it be more right to say that '... each N-bias profile is displayed as a function of an adjusted height being the height minus the derived PBLH.'?**
    - The authors feel that 'normalized' is an acceptable word to use; however, the reviewer's suggestion offers more clarity.
    - Removed 'normalized' and added '…displayed as a function of an adjusted height, which is the height minus the derived…'
    - Change made on line 206 and line 321 (line 324 in track changes version)

- line 218: **Remove the minus in front of '120' or skip the 'E' (don't have both). The same can be said about the longitudes given in sections 3.3.2 and 3.3.3. Please revise.**
    - Directional notations have been removed from all latitude and longitude references and the authors will maintain consistent use of positive latitude (North) and longitude (East) and negative latitude (South) and longitude (West).

- line 236: **Perhaps here '... MAGIC radiosondes (rds) ...' to be consistent with Fig. 4.**
    - Thank you for catching this, '(rds)' has been added for consistency.

- line 243: **Add 'the' in front of 'gradient method'.**
    - Added 'the' in front of 'gradient method'.

- line 246: **Comma instead of period before 'while'.**
    - The period has been changed to a comma.

- lines 246-247: **Please support with references regarding trade-wind inversion and mixing layer.**
    - Added citations, "Ao et al., 2012; Xie et al., 2012; Riehl, 1979" as references for use of 'ducting layer' and 'trade-wind inversion'
    - Added citation, "Xie et al., 2006" as reference to 'mixing layer'

- lines 249-250: **Refer to section 2.1 instead of repeating numbers.**
    - Text from line 247 beginning with "Note the radiosonde…" to line 250, ending with "…within 2-3 km" has been deleted and replaced with the following:
        - "Due to the difference in vertical sampling noted in Section 2.1, the ERA5…"

- line 276: **"All parameters are interpolated to a 10 m vertical grid." How do you interpolate these parameters? They are not functions of height. I'm not sure this new added sentence makes any sense.**
    - The reviewer is correct; the sentence does not make any sense. The reference was meant to notify the reader that the mentioned parameters were calculated from interpolated refractivity profiles.
    - The sentence has been removed.

- line 279: **Should 'is' be 'as'?**
    - Yes. Changed 'is' to 'as'.

- line 302: **'bars' and 'bar'. Please be consistent.**
  - Changed 'bar' to 'bars'.

- line 347: **Could it be just "peak height" instead of "its peak N-bias occurring height"?**
  - Yes. "N-bias occurring" has been removed.

- line 350: **Comma instead of period before 'whereas'.**
  - Punctuation has been changed.

- line 353: **Please improve the syntax of this sentence: "Overall, the N-bias in ERA5 are smaller than radiosonde in all bins."**
  - Syntax has been improved to the following:
  - "Overall, the *N*-bias values for the ERA5 data set are less than the *N*-bias values calculated from the radiosonde data set for each longitude bin."

- line 357: **"Note that the ...". However, it is not possible to see the numbers from the figure. Perhaps just skip "Note that".**
  - "Note that" has been removed from line and replaced with "The".

- line 358 **(and other places): 'PBL height' -> 'PBLH'.**
  - All references to PBL height have been changed to PBLH except where first defined in the Abstract (line 13) and main text of the introduction (line 33).

- line 359: **I think I know what you want to say, but this does not make sense: "... shows greater difference than the height of peak N-bias ...". Please revise.**
  - Paragraph from lines 357 to 361 has been revised to the following:
  - "Note that the PBLH is above the height of the peak *N*-bias for both data sets. The MAGIC data shows a maximum difference of 100 m (−137.5°) and a minimum difference of ~15 m (−152.5°) while the ERA5 PBL height shows greater values for maximum difference (230 m at −142.5°) and minimum (45 m at −157.5°)."

- lines 393-394: **Numbers here are still wrong/not updated compared to table 1. However, it seems that they do correspond to the curves in Fig. 9a. For example, the easternmost points in Fig. 9a are slightly above 6 (blue) and 8 (red), whereas the median numbers in Table 1 are slightly below 6 and 8, respectively. Please verify that the numbers in Table 1, in the text, and in the figures are correct and consistent. Please also indicate that the numbers in Table 1 are in percent (I assume they are).**
  - The reference "Fig. 9a and Table 1" on line 394 should not have included Table 1. The reviewer is correct in noting that the numbers reference points in Figure 9.
  - Table 1 is a supplement to Figure 8 and should not have been noted here. Thank you for bringing this to the author's attention.
  - The reference to Table 1 has been removed from line 394.

o The numbers listed Table 1 are percentages and referenced correctly from lines 339-360.
o Table 1 has been updated for clarity, the caption for Table 1 has been updated as well.

- lines 409-411: **A revised sentence here reads: "The reason for the lower correlation value in MAGIC data is attributed to outlier cases when the radiosonde N-bias profiles with a double peak at which the larger magnitude bias is located" This is grammatically wrong. Please revise.**
  o In response to major point #3, section 3.3.3 has been removed.

---

## Author Response (AR3)

**Author Responses to Reviewer Comments**

**Third Round of Peer Review**

**We greatly appreciate the constructive comments from the reviewer and the editor. We have carefully considered all the comments and have provided detailed and thoughtful responses to each concern.**

**Major revision**

**Editor comments on Report #1 dated 2024-05-5**
The issue raised by the Reviewer needs to be properly addressed in an updated version.

**Reviewer Comments from Report Dated 2024-05-05**
I am happy to see a much-improved manuscript. However, there are still some issues that needs to be addressed before I can recommend publication.

**We appreciate the reviewer's comments on the interpolation issue and its potential impact on the scientific results of the paper.**

**The authors would like to reemphasize that the major results of this paper focus on the high-resolution radiosonde data. The interpolation method issue raised by the reviewer does not affect the radiosonde data analysis, but may affect some of the ERA5 data analysis as the reviewer pointed out. We have clarified the point that some results related to ERA5 data analysis could be affected by the interpolation issue which we hope addresses the reviewer's concerns in addition to further examination included in our responses. Please see below for our detailed responses.**

1) Figure 2d (and its implications) is still a major problem in my opinion. The authors have now made clear that they use quadratic interpolation (now written in the text), and that the calculation of the derivative also uses quadratic interpolation (in their answer to me). This means that the second derivative becomes constant in intervals (and is equal to the first derivative of the gradient). From their explanation, I think I can understand why this procedure can result in both short (about 20 m) and longer (about the distance between model levels) intervals with constant first derivatives of the gradient, which appear as the sawtooth-like features. In the text the authors now write: "Note that the weak gradients seen above the minimum in the ERA5 refractivity gradient (Fig. 2d) are a result of the vertical derivative being calculated from the interpolated ERA5 refractivity profile and do not appear for larger interpolation intervals suggesting that the non-linearity of the ERA5 vertical grid at this height affects the vertical gradient." I don't understand why the non-linearity of the ERA5 vertical grid gets the blame, I think it comes from the interpolation method of using only quadratic interpolation (cubic interpolation would have been a better choice). In any case, the sawtooth-like features in Figure 2d are not only the weak ones above the minimum gradient. The minimum gradient itself has a short interval right above (about 20 m), and a longer interval right below (about 150 m) with

constant first derivatives. As I see it, this is a sawtooth-like feature right at the minimum gradient.

    a. **We apologize for the confusion in our previous response. The "non-linearity of the ERA5 vertical grid" refers to the uneven vertical sampling intervals of the ERA5 data, i.e., much larger sampling interval at higher altitude, which is contrary to the relatively constant sampling interval (~ 8 m) in radiosonde data.**

    b. **Note that the raw vertical sampling interval of ERA5 refractivity profile of ~ 160-200m between 2 km and 3 km. The refractivity profile is interpolated into 10 m intervals with three different interpolation methods (*linear, quadratic, and cubic spline*). Figure RS1 illustrates the original ERA5 refractivity profile (used in Figure 2 of the manuscript) and the interpolated profiles using each method (a and c), as well as the resulting refractivity gradient profiles (b and d). All three interpolation schemes lead to identical refractivity profiles. The "sawtooth-like features" in the gradient plot above and below the peak refractivity gradient at ~2.1 km are also evident in both linear and cubic-spline interpolation methods, in addition to the quadratic interpolation method used in this study. Detailed differences are more explicitly seen in the enlarged figure (right panels of Fig. RS1).**

    c. **On the other hand, Figure RS2 demonstrates the same three interpolation methods applied on the colocated MAGIC radiosonde refractivity profile (native vertical sampling of ~8 m). The quadratic interpolation is almost identical to cubic-spline interpolation. This confirms that the high-resolution radiosonde data is not sensitive to the selection of the high-order interpolation schemes (quadratic or cubic).**

    d. **Therefore, such fine "sawtooth-like features" are the results of the interpolation when the interpolation interval is much smaller than the vertical sampling interval of the original data.**

    e. **Based on these results, the authors believe that a full reassessment of the data with a new interpolation method will not improve the results of this study.**

[Figure]

[Figure]

**Figure RS1. (Left 2 panels): Comparison of** ERA5 **(a) refractivity and (b) vertical refractivity gradient profiles from Figure 2 in the original manuscript. Note the ERA5 has ~160-200 m vertical sampling interval between 2 km and 3 km. In both panels, raw ERA5 data (solid black) and 10 m interpolated profiles using linear (blue), quadratic (gold) and cubic spline (purple) interpolation schemes in IDL. (Right 2 panels): the enlarged portion of the left panels from 2.2 km to 2.8 km, to illustrate the gradient difference for three interpolation schemes.**

[Figure]

**Figure RS2. Same as Figure RS1 above, but for the colocated** MAGIC radiosonde **(RDS) (a) refractivity and (b) refractivity gradient profile. Note the radiosonde profile has the raw vertical sampling interval of ~ 8 meters from surface up to ~30 km.**

2)  Further the authors write: "These features of approximately 15 N-units/km magnitude are only noticed in the plotting and do not impact the results of the study, as only the moisture-induced minimum gradient values are large enough in magnitude to exceed the minimum gradient threshold." I don't understand why the authors say that the features are only noticed in the figure. I assume that the figure shows the data, so the sawtooth-like features are in the data (due to the quadratic interpolation), not only in the figure. Maybe they mean that these minor gradients are not affecting the estimates of minimum gradient and height. However, the largest sawtooth-like feature right at the minimum gradient is more than 200 N-units/km. It may be that quadratic interpolation do not affect the estimate of the magnitude and height of the minimum gradient much, but that at least is unclear.

    a.  **Sorry for the confusion. Yes, we believe that the quadratic interpolation does not affect the estimate of the magnitude and height of the minimum gradient, as the sawtooth feature is higher order fine structure in the gradient profile.**

3)  On the other hand, I do think the method (creating these sawtooth-like features) may affect the estimate of the ducting thickness (see below). I strongly suggest that the study is done with a better interpolation method. If this is not done, then at least the authors should write in the text and in the conclusion that the results may be affected by the interpolation method and by how the gradients are calculated. Now that I think I understand how the sawtooth-like features appear, I am even more concerned that the results in Figure 6 and elsewhere are affected. It seems clear to me that the narrow (about 20 m) intervals of constant first derivatives near the minimum gradients are artificial, and a result of the interpolation method, and that this can affect the estimated ducting

thickness. In their answer the authors write: "As an example, an ERA5 profile that is not interpolated can have a minimum gradient value of -160 N-units/km and if the raw profile resolution at that point is 50 m, one could argue the ducting layer thickness is 50 m. However, a 50 m ducting layer thickness could be just as artificial as the reviewer is claiming for a 10 m thickness.". I disagree. The 50 m ducting layer thickness (if so estimated) would be what ERA5 suggests. It may not be the actual value of the ducting layer thickness in the real atmosphere, but I think that is irrelevant here. Due to the coarser resolution ERA5 profile, the 50 m thickness (if so estimated) would be the right number. Thicknesses of only 10 m, if they are a result of the interpolation method, would be wrong. Again, I urge the authors to repeat the study with a better (higher order) interpolation method, or clearly write that the results may be affected by the interpolation method and by how the gradients are calculated.

a. **The authors understand the reasoning for the reviewer's strong opinion on the interpolation method. However, changing the interpolation method would not remove the presence of the "saw-tooth like features" as seen in Figures RS1 and RS2. As stated previously, the issue arises from calculating the gradient of a profile that was interpolated to a higher resolution than the raw data. To remove the possibility of this occurrence, the resolution of the interpolated profile would have to be decreased to the coarsest vertical resolution from the ERA5 data in the lowest 3 km, which would be approximately 200 m. If this were the case, then there would be no point to using the radiosonde data as it would need to be interpolated to the same resolution for a reasonable comparison.**

b. **We agree with the reviewer that quadratic interpolation could introduce higher-order fine structure in the gradient plot right next to the sharp gradient (e.g., the PBL height). However, we believe that the PBL height detection is not affected.**

c. **As stated in the response for reviewer comment #2, such fine "sawtooth-like features" are the results of the interpolation when the interpolation interval is much smaller than the original data vertical sampling (e.g., ERA5). The authors believe that a full reassessment of the study with a new interpolation method will not improve the results of this study.**

d. **The authors believe the research is sound and the findings are of note, especially since the MAGIC data are not subject to the shortcomings of the interpolated ERA5 data. The authors believe that the methods of analysis used on the ERA5 data in this paper will also provide an example of the differences when high resolution model data are interpolated and compared to high resolution radiosonde data. We feel that a new iteration of the study with updated methods of both interpolation and calculation of the gradient would not offer the solution the reviewer suggests; however, we have no issue with giving full disclosure of the methods used and acknowledgment that the current methods may impact some of the results for the ERA5 data**

i. **Section 2.3, lines 174-180, "It should be noted that the weak "saw tooth-like" gradients seen above the minimum in the ERA5 refractivity gradient (Fig. 2d) are a result of the vertical derivative being calculated from the interpolated ERA5 refractivity profile. When interpolating the relatively coarse vertical resolution ERA5 profile (up to 200 m in the lowest 3 km) into 10 m vertical sampling, the higher-order interpolation could lead to fine structure in the first order derivative. However, these minor gradients do not affect the estimates of minimum gradient and associated heights."**

ii. **Section 4, lines 447-449, "Further, the ERA5 results may be affected by the interpolation resolution and gradient calculation. Both warrant a more comprehensive study in the future."**

e. **Thank you for the comment acknowledging that the author's intention to state that the minimum gradient used to identify the PBLH is not affected by these weak gradients was unclear. The reviewer's sentence stating this explicitly has been added at lines 177-178.**

4) The authors followed my suggestion and removed section 3.3.3. Because of that, they also removed equation 2, which means that the sharpness parameter is no longer defined. However, the sharpness parameter is still discussed and shown in figures (Fig. 4c). Maybe equation 2 just needs to be reintroduced. Please revise.

a. **Thank you for pointing this out. The paragraph that introduces the sharpness parameter has been reintroduced (lines 138-143) as follows:**

i. **"To assess the robustness of the PBLH detection with the gradient method, Ao et al. (2012) introduced the sharpness parameter ($\widetilde{N}'$) to measure the relative magnitude of the minimum gradient, which is defined as the ratio of the minimum vertical refractivity gradient ($N'_{min}$) to the root mean square ($N'_{RMS}$) of the refractivity gradient profile from surface to 5 km as follows:**

$$\widetilde{N}' \equiv -\frac{N'_{min}}{N'_{RMS}}, \qquad\qquad (2)"$$

5) line 162: 'N-units/10' is still not the right unit. I believe it should be '10 x N-units' or deca N-units (just like km = 1000 m; when using km instead of m it reduces the numbers on the axes by a factor of 1000). However, here units for the other variables are not specified, so why for refractivity? - one could just say (N) here. Please see my comments on earlier revisions about the N-unit issue in Figure 2 and revise.

a. **The units on the x-axis of Figure 2 have been revised according to the reviewer's suggestion and the text has been modified accordingly.**

6) Although the authors have now removed the reference to Table 1 when discussing Fig. 9, the numbers in Table 1 are still the same and thus not consistent with Fig 9. I don't understand that. Why are the numbers for the peak N-bias in Table 1 different from the peak N-bias in Fig. 9? If the numbers are supposed to be different (in which case I misunderstood something), then please explain in the text why they are different. Otherwise please revise either Table 1 or Fig. 9 (and the numbers in the text).

a. The numbers in Table 1 are intended to accompany Figure 8. Clarification has been added to line 351-352 with the following text.
      i. "Table 1 lists detailed statistics of the peak *N*-bias values at each bin for both radiosonde and ERA5 data seen in Fig. 8".
   b. The numbers in Table 1 are included in the discussion of Figure 8 (lines 355-359).
7) line 120: non-linear -> non-equidistant.
   a. Text has been changed from "non-linear" to "non-equidistant".
8) line 357: looks like it is more than 15 m in Fig 8.
   a. The reviewer's observation is correct, and the numbers have been updated in lines 363-365 with the following text:
      i. "…maximum difference of 100 m (−157.5°) and a minimum difference of ~70 m (−142.5°) while the ERA5 PBLH shows greater values for maximum difference (140 m at −132.5°) and minimum difference (60 m at −157.5°)."
   b. The reason for the change is that upon checking the reviewer's noted inconsistency it was discovered that the data describing the difference between the PBLH and height of the peak *N*-bias were from a previous iteration of this calculation and were not consistent with the values depicted in Figure 8. This has been corrected in the text as described above.
9) line 358: of -> difference.
   a. Text has been corrected from "minimum of" to "minimum difference".
10) line 419: approximately 80 meters below -> slightly below (also in abstract). The 80 meters came out of Section 3.3.3, which has been removed.
   a. The text "approximately 80 meters below" has been changed to "slightly below" in both locations so there is no longer reference to a section that has been removed from the document.

**Editing Comments from Response Dated 2024-03-27**

**Notification to the authors**:
Please ensure that the colour schemes used in your maps and charts allow readers with colour vision deficiencies to correctly interpret your findings. Please check your figures using the Coblis – Color Blindness Simulator (https://www.color-blindness.com/coblis-color-blindness-simulator/) and revise the colour schemes accordingly. => Figs. 2 and 6

- **Figure 2: The vertical profile lines are depicted in separate textures as noted in text, in caption, and the legend and they are consistent for all four panels within the figure. Additionally, the texture of the vertical gradient profiles in panels b and d correspond to the original variables in panels a and c, respectively. The ducting layers are described in a similar fashion with the top and bottom layer textured with dotted lines and the height of the minimum gradient identified with a dashed line. The green dashed line has been changed to gold as it is more distinguishable when viewed with each filter from the Coblis tool.**
- **Figure 6: The figure has been altered to use colors more agreeable to color vision deficiencies (red, gold, blue, purple). Additionally, instead of all open circles, each**

bin uses a different character (circle, square, diamond, asterisk, plus sign, triangle, and X). These combinations are identifiable using all filters provided in the Coblis tool. References in text and captions have been altered to reflect the change of this figure as well.

---

## Author Response (AR4)

**Author Responses to Editor Comments**

**Second Round of Editorial Review**

**We extend our gratefulness for such a thorough review process and welcome each round of peer and editorial review and the associated changes as a significant improvement of this manuscript. Additionally, we thank the editor for the suggested minor revisions. Please see the acknowledgment and associated changes/comments in bold below.**

**Minor revision**

**Public justification (visible to the public if the article is accepted and published)**:
Before publication the following should be done: A few editorial corrections and reformulations regarding the results should be implemented.

1) There are some Track Change marks visible in the manuscript in lines 132-143. Please check (and correct).

- **Thank you for correcting this mistake, all track change errors have been removed.**

2) Regarding the possible impact of "saw-tooth" structures:
I suggest to write "only marginally" instead of "do not" in line 179.

- **The verbiage "do not" has been replaced with "only marginally" on line 179.**
   - **The sentence now reads from line 179-181, "However, these minor gradients only marginally affect the estimates of minimum gradient and associated heights from ERA5 and is most often overshadowed by the PBLH gradient."**

3) Regarding the use of "daN-units".
Please correct the following:
Fig.2a, Fig.2c legend: Change "daN-units" to "N"
Fig.2a, Fig.2c x-axis: Change "daN" to "N"
Fig.2 caption: Change "daN-units" to "N"

- **Changes have been made to the caption, legend and X-axis of Figure 2.**
- **Correction was also made on line 168 by changing "daN-units" to "N in daN-units".**

4) Regarding the thickness of the ducting layers:
I suggest to add the following sentence in Section 3.2:
"It should be noted that the estimated thicknesses of the ducting layers, especially for ERA5, may be affected by the chosen interpolation method."

- **We agree with the suggestion of the editor, and the line "It should be noted that the estimated thicknesses of the ducting layers, especially for ERA5, may be affected by the chosen interpolation method." has been added to lines 295-296.**

5) Table 1 and Fig. 9 do not seem to be consistent.
Here is one example:
Table 1, RDS median, 122.5 degrees: -7.86
Fig.9a, at 122.5 degrees: the red value is : -8. … (-8.something)
Several values are inconsistent.
Please check, and correct.

- **Table 1 is meant as a supplement for Figure 8. The reason is that Figure 8 is a cross-section with longitude represented on the X-axis. Additionally, a secondary axis representing a repeating scale of -5, 0, and 5 for each profile across the transect was incredibly complicated and cluttered the scale to a point of being illegible. As a result, Table 1 was added to the document directly after Figure 8 in order to illustrate the numerical values (median and M.A.D.) for the profiles in Figure 8. Table 1 is stated as a reference for Figure 8 in the caption (of Figure 8) as well as on line 355 in the main text.**
- **As an update to avoid and confusion to the reader, a reference to Figure 8 has also been added in the caption of Table 1 with the following text:**
  - **"Table 1: Peak values of median *N*-bias and corresponding MAD (%) values for MAGIC radiosondes (RDS) and ERA5 for each 5° bin seen in Figure 8."**

Please ensure that the colour schemes used in your maps and charts allow readers with colour vision deficiencies to correctly interpret your findings. Please check your figures using the Coblis – Color Blindness Simulator (https://www.color-blindness.com/coblis-color-blindness-simulator/) and revise the colour schemes accordingly. => Figs. 2 and 6

- **Figure 2: The vertical profile lines are depicted in separate textures as noted in text, in caption, and the legend and they are consistent for all four panels within the figure. Additionally, the texture of the vertical gradient profiles in panels b and d correspond to the original variables in panels a and c, respectively. The ducting layers are described in a similar fashion with the top and bottom layer textured with dotted lines and the height of the minimum gradient identified with a dashed line. The green dashed line was changed to gold as it is more distinguishable when viewed with each filter from the Coblis tool.**
- **Figure 6: The figure was altered to use colors more agreeable to color vision deficiencies (red, gold, blue, purple). Additionally, instead of all open circles, each bin uses a different character (circle, square, diamond, asterisk, plus sign, triangle, and X). These combinations are identifiable using all filters provided in the Coblis tool. References in text and captions were altered to reflect the change of this figure as well.**